# Adult neurogenesis improves spatial information encoding in the mouse hippocampus

M. Agustina Frechou [1,2,7], Sunaina S. Martin [1,2,8], Kelsey D. McDermott [1,2], Evan A. Huaman[1,2], Şölen Gökhan[3], Wolfgang A. Tomé[3,4], Ruben Coen-Cagli [1,5,6] & J. Tiago Gonçalves [1,2] ✉

Adult neurogenesis is a unique form of neuronal plasticity in which newly generated neurons are integrated into the adult dentate gyrus in a process that is modulated by environmental stimuli. Adult-born neurons can contribute to spatial memory, but it is unknown whether they alter neural representations of space in the hippocampus. Using in vivo two-photon calcium imaging, we find that male and female mice previously housed in an enriched environment, which triggers an increase in neurogenesis, have increased spatial information encoding in the dentate gyrus. Ablating adult neurogenesis blocks the effect of enrichment and lowers spatial information, as does the chemogenetic silencing of adult-born neurons. Both ablating neurogenesis and silencing adult-born neurons decreases the calcium activity of dentate gyrus neurons, resulting in a decreased amplitude of place-specific responses. These findings are in contrast with previous studies that suggested a predominantly inhibitory action for adult-born neurons. We propose that adult neurogenesis improves representations of space by increasing the gain of dentate gyrus neurons and thereby improving their ability to tune to spatial features. This mechanism may mediate the beneficial effects of environmental enrichment on spatial learning and memory.

Environmental factors such as cognitive stimulation and exercise can lead to memory improvements and are associated with a lower incidence of aging-related cognitive decline[1,2], as well as protection against neurodegeneration[3–5]. While the effects of these environmental manipulations are complex and systemic, in the mouse hippocampal dentate gyrus (DG) environmental enrichment increases adult neurogenesis[6], resulting in the addition of adult-born neurons (ABNs) that contribute to DG-mediated memory[7]. Several studies have found

that increasing ABN numbers enhances memory and cognition. ABNs have been found to contribute to diverse cognitive tasks such as spatial memory[8], cognitive flexibility[9], discrimination between memories of similar events[10–13], and memory elimination/forgetting[14]. However, the field still lacks a unified conceptual framework for how ABNs influence hippocampal function to mediate these different behavioral phenotypes. In mice, ABNs primarily contribute to DG-dependent memory, during a critical period at 4–6 weeks post mitosis[15,16], when the still-

[1]Dominick P. Purpura Department of Neuroscience, Albert Einstein College of Medicine, Bronx, NY, USA. [2]Gottesmann Institute for Stem Cell Biology and Regenerative Medicine, Albert Einstein College of Medicine, Bronx, NY, USA. [3]Saul R. Korey Department of Neurology, Albert Einstein College of Medicine, Bronx, NY, USA. [4]Department of Radiation Oncology, Albert Einstein College of Medicine, Bronx, NY, USA. [5]Department of Systems and Computational Biology, Albert Einstein College of Medicine, Bronx, NY, USA. [6]Department of Ophthalmology and Visual Sciences, Albert Einstein College of Medicine, Bronx, NY, USA. [7]Present address: Laboratory of Neurotechnology and Biophysics, The Rockefeller University, New York, NY, USA. [8]Present address: Department of Psychology, University of California San Diego, La Jolla, CA, USA. ✉e-mail: tiago.goncalves@einsteinmed.edu

immature neurons are already synaptically integrated into DG circuits but remain more plastic and excitable than mature DG granule neurons[17,18]. Nevertheless, it is unclear how these properties change DG circuits to improve memory.

The DG and other areas of the hippocampus play an essential role in episodic and spatial memory[19,20] by forming a 'cognitive map'—a neural representation of space to which objects and events can be bound. DG neurons achieve this in part by functioning as 'place cells', firing selectively to a single location in space[21,22]. But even neurons that are not selectively tuned to a single place can contribute to the spatial information encoded in neural populations. Neuronal activity recordings from large populations can be used to decode the position of an animal within an environment[23,24], with the decoding accuracy reflecting the spatial information content. These neural representations of space are essential for spatial memory, as was recently demonstrated[25]. Since ABNs contribute to spatial memory, one might expect that these cells would be sparsely active and finely tuned to a specific place. However, immature ABNs are highly excitable[15,17,26–28] and less spatially tuned than their mature granule cell counterparts[28,29]. Current models of DG function propose that ABNs contribute to memory through the activation of local interneurons, resulting in a net increase in inhibition[30–34]. Yet, it is unclear how this would affect neural representations of space during the formation of a new memory, as the animal explores a novel environment. To address these open questions, we investigated whether increasing adult neurogenesis by housing mice in an enriched environment (EE) would change the DG spatial code. We recorded activity from mature granule neurons in the DG using two-photon calcium imaging, as the mice walked on a moving treadmill with spatial cues. To directly demonstrate the role of adult neurogenesis in the DG neural code, we either chronically ablated DG neurogenesis by focal irradiation of the hippocampus, or acutely silenced immature ABNs using chemogenetics, and assessed the effect of these manipulations on DG spatial information content. We found that EE-exposure results in increased spatial information content in the DG, and that this increase requires immature ABNs. Moreover, our results show that ABNs act to increase the mean activity rates of individual mature DG granule neurons, and the spatially selective activity of neurons that are tuned to a single location on the treadmill. These findings elucidate the action of ABNs on the encoding of information in the DG.

## Results

### Adult neurogenesis increases DG spatial information in animals exposed to EE

Environmental enrichment and exercise have been found to improve cognitive performance, including spatial memory in humans[2] and rodents[6], as well as increase the number of ABNs in the DG. We asked whether this improvement in spatial memory could be due to improved neural representations of space in the hippocampus, and whether adult neurogenesis is required to mediate these changes. To address these questions, we imaged the granule cell layer of the DG of mice housed in regular cages (RC) or in an EE with running wheels (Fig. 1a). Two groups of animals had DG neurogenesis ablated by prior bilateral focal irradiation using opposing large fields (Irr.+RC, Irr.+EE, Supplementary Fig. 1a, b)[35], while the other two were not irradiated (RC, EE). Irradiation results in the permanent ablation of DG adult neurogenesis, while preserving subventricular zone / olfactory bulb (OB) neurogenesis, as confirmed post hoc by the persistence of DCX immune-reactive cells in the OB (Supplementary Fig. 1c). We labeled DG neurons of 10 weeks-old C57Bl6/J mice with an AAV encoding the red calcium sensor jRGECO1a[36] and implanted them with a titanium imaging 'window' (Fig. 1a), resting above the surface of CA1 in the dorsal hippocampus[37]. This approach labels mature DG neurons, but no immature ABNs (Supplementary Fig. 2), since both intermediate progenitor cells and early post-mitotic ABNs are sensitive to AAVs and

will undergo apoptosis upon infection[38]. After the second week post-mitosis, ABNs develop tolerance to AAV infection, however this cohort will have matured by the time of imaging, past the 4-6 week critical period when ABNs are thought to have their maximum impact on cognition[15,16]. Of note, the AAV injection results in a reduction of the number of ABNs in high-expression areas next to the epicenter of viral injection, which is also the region that was imaged in our experiments (Supplementary Fig. 2a). After the implantation surgery, mice were housed either in RC conditions for 3.5 weeks or in EE conditions for 2 weeks followed by 1.5 weeks in a RC. Enriched environments with running wheels increase neuronal proliferation and survival, therefore acting on a relatively wide range of progenitors/neurons. However, their largest effect is an increase in survival of neurons that are ~ 7 to 21 days post-mitosis[39]. Therefore the bulk of the unlabeled, EE-enhanced ABN population will be ~ 4.5–6.5 weeks old at imaging[39]. A single calcium imaging session took place ~ 3.5 weeks after surgery (Fig. 1a, b). Mice were head-fixed to a treadmill with a belt composed of different texture materials (Fig. 1a), and imaged unanesthetized as the treadmill was rotated at a mean speed of ~420 cm/min (Supplementary Fig. 3a, b, Supplementary Movie 1). DG neuronal activity was sparse with an average of 86 neurons active during a 9 min recording session in a field of view of up to 343 µm x 343 µm, corresponding to 4.3 ± 0.8% of all labeled putative neurons (Fig. 1b). As expected, post-mortem histological analysis showed that DG adult neurogenesis was permanently ablated in irradiated animals, while EE animals showed a ~ 2-fold increase in doublecortin (DCX) positive neurons relative to their RC counterparts (Fig. 1c, d). To estimate spatial information content from the imaged neurons we trained a linear decoder to decode the position of the mouse on the treadmill from the calcium traces (Fig. 1e, trained on 75% and tested on 25% of the data). Since numbers of imaged and active neurons were different across mice, we sub-sampled our datasets to compare decoding performance of equally-sized populations across different mice. Exposure to EE resulted in an increase in decoding accuracy compared to mice in RC (Fig. 1f), reflecting an increase in spatial information content in the DG. While multiple exposures to a single context are known to increase hippocampal spatial information within that same context[40,41], our findings show that 2-weeks of EE housing can increase spatial information in future exposures to a novel context. In addition, EE exposure failed to increase decoding accuracy in irradiated mice, in contrast with non-irradiated mice. Irradiated animals also had lower decoder accuracy than their non-irradiated counterparts. These results suggest that the effects of EE on spatial encoding in the DG are driven primarily by immature ABNs, and that decreasing adult neurogenesis results in a decrease in spatial information in the DG, which could explain why reducing neurogenesis results in memory encoding deficits.

### Ablating adult neurogenesis decreases single-cell spatial information

Next, we studied whether changes in spatial information in the DG were driven by changes at the population or single-cell level. The information content of neural populations is determined jointly by the response properties of individual neurons, i.e. their tuning and the variability of their activity when repeatedly presented with the same stimulus (in this case each location on the treadmill belt) and by the structure of noise correlations[42] (NCs), i.e how this variability is correlated among pairs of neurons[43,44].

We quantified spatial information using Population Fisher Information (a measure of population sensitivity to spatial location, equivalent to d-prime (d'))[45] – see methods) because it allows us to assess population-wide effects more accurately than decoding performance[46] and because it allows us to express the sensitivity also in terms of discrimination thresholds. First, to test the contribution of noise correlations to spatial information encoding in the DG, we

created surrogate datasets where we shuffled the trial order independently for each neuron. The response of each neuron is given by the response at the same location but on a different trial/timepoint. Therefore, noise correlations are destroyed, but the tuning of individual neurons is preserved. Our results show that spatial information in the shuffled datasets was slightly increased, although this difference was not significant (Fig. 2a–d), indicating that noise correlations have no effect on decoding accuracy in the DG or may be even detrimental. This result suggests that the effect of ABNs on spatial information cannot be explained by a qualitative change in the structure of noise correlations from detrimental to helpful for coding (see Discussion for

the relation with two other recent studies of how noise correlations impact spatial information in DG and CA1[47,48].

Next, we calculated the spatial information of individual neurons (Fig. 2e). We found that ablating neurogenesis strongly reduced spatial information of single neurons and that this reduction was present both among the neurons with the most spatial information as well as neurons that had lower spatial encoding (Fig. 2f). As a more intuitive measurement, we also calculated a discrimination threshold, that is the minimum distance between two points on the treadmill that will be discriminated correctly 70% of the time (lower thresholds reflect a more precise spatial code; Fig. 2g). As expected, irradiated cohorts

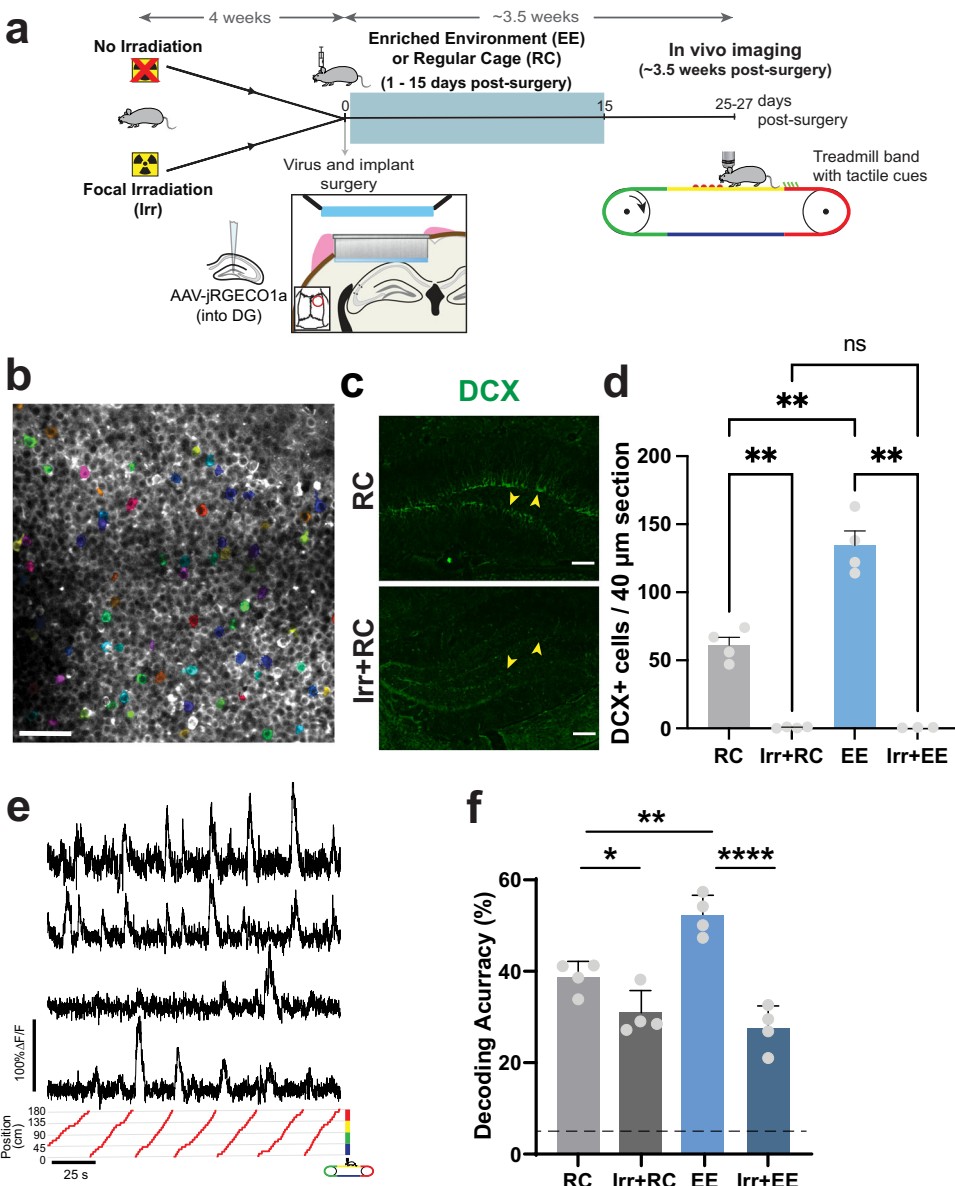

**Fig. 1 | Environmental enrichment increases spatial information encoding in the DG but this effect is blocked in mice with ablated adult neurogenesis.**
**a** Experimental timeline, including irradiation, surgery, housing environment, and in vivo imaging. **b** Example of calcium imaging field of view. Cells that were active during a recording are shaded in color. **c** Immunofluorescence labelling of DCX positive neurons in the DG of non-irradiated (top) and irradiated (bottom) mice. Arrows denote neurogenic subgranular layer in the upper and lower leaves of the DG where DCX-expressing cells can be found. **d** Number of DCX-expressing neurons in imaged mice from regular cage (RC), enriched environment (EE) groups and corresponding irradiated groups (Irr+RC, Irr+EE). ($n_{RC}$ = 4 mice, $n_{Irr+RC}$ = 4 mice,

$n_{EE}$ = 4 mice, $n_{Irr+EE}$ = 3 mice, average of three 40 μm slices per mouse, Welch ANOVA test with Dunnett's T3 multiple comparisons, RC vs Irr+RC **$p$ = 0.008, RC vs EE **$p$ = 0.009, Irr+RC vs Irr+EE ns $p$ > 0.999, EE vs Irr+EE **$p$ = 0.004). **e** Example calcium traces (top) and respective position of animal on the treadmill (bottom). **f** Accuracy in decoding position of mouse on treadmill from calcium traces (RC vs Irr+RC: *$p$ = 0.006, EE vs Irr+EE: ****$p$ = 2.15 × 10$^{-5}$, **RC vs EE: $p$ = 0.03, ANOVA, Holm-Sidak correction for multiple comparisons, $n$ = 4 mice, 42 neurons subsampled per mouse. Dotted line is chance performance level (5%). Error bars represent +/- SEM in all plots. All scale bars = 50 μm.

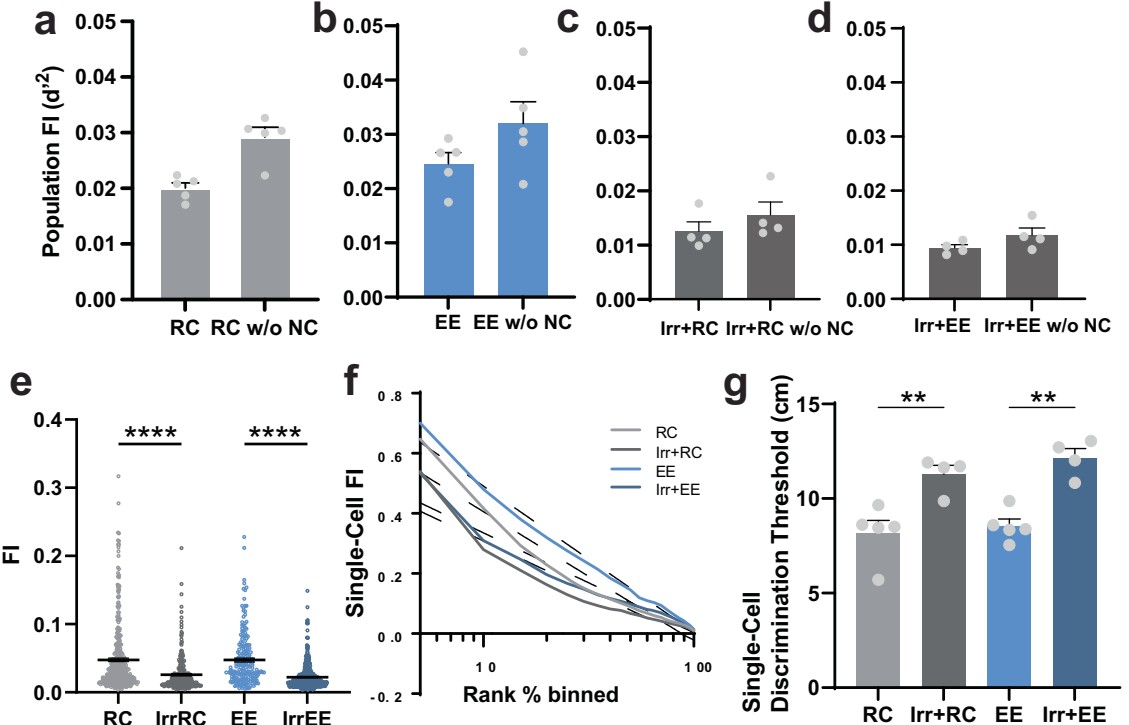

**Fig. 2 | Ablating adult neurogenesis decreases spatial information content at the single cell level. a–d** Population Fisher information determined with noise correlations and after random shuffling to disrupt noise correlations (RC vs RC w/o NC: $p = 0.057$ n.s., $n = 5$ mice per group, EE vs EE w/o NC: $p = 0.15$ n.s., $n = 5$ mice per group, Irr+RC vs Irr+RC w/o NC: $p = 0.20$ n.s., $n = 4$ mice per group, Irr+EE vs Irr+EE w/o NC: $p = 0.014$, $n = 4$ mice per group, Mann-Whitney U test). **e** Single-cell spatial information content determined using Fisher information (FI) (RC vs Irr+RC: ****$p < 1/100000$, $n_{RC} = 5$ mice, 277 neurons, $n_{Irr+RC} = 4$ mice, 253 neurons, EE vs Irr +EE: ****$p < 1/100000$, $n_{EE} = 5$ mice, 201 neurons, $n_{Irr+EE} = 4$ mice, 412 neurons, statistical analysis: bootstrap (two-sided) and Bonferroni correction for multiple comparisons (see methods). Error bars are mean ± SEM. **f** Distribution of spatial information content across the imaged neurons. **g** Distance between two positions that DG single-cells are able, on average, to discriminate correctly 70% of the time (RC vs Irr+RC: **$p = 0.0038$ $n_{RC} = 5$ mice, $n_{Irr+RC} = 4$ mice, EE vs Irr+EE: **$p = 0.0012$, $n_{EE} = 5$ mice, $n_{Irr+EE} = 4$ mice, statistical analysis: ANOVA, Holm-Sidak correction for multiple comparisons). Error bars represent +/- SEM in all plots.

both in RC and EE conditions had increased discrimination thresholds. These findings indicate that the observed changes in DG spatial information occur primarily at the single-cell level, which led us to further study how the activity of individual cells changes with EE exposure and the ablation of adult neurogenesis.

## Ablating adult neurogenesis reduces single cell activity and tuning

To assess how single cell spatial information changes in the DG, we mapped the calcium activity traces to the position of the mouse on the treadmill to generate a tuning vector (Fig. 3a, b). The mean calcium signal at each location was computed for every neuron and normalized to the time the mouse spent at that position. The modulus of the tuning vector was used as a spatial tuning index that reflects the place-specificity of activity[29]. Irradiated cohorts had significantly lower mean tuning indices than non-irradiated animals (Fig. 3c). While both irradiated and non-irradiated groups had highly tuned cells, the distribution of tuning indices skewed lower in the irradiated cohorts, both for highly tuned cells as well as their low-tuning counterparts (Fig. S4a), suggesting that the lower information content in these mice could result from a decrease in the place-specificity of neuronal activity. This decrease in tuning could be caused by an overall change in the calcium activity levels of DG neurons or by changes in how this activity is distributed along the treadmill belt. Since adult neurogenesis is thought to contribute to a net decrease in DG activity through the increased activation of feedback inhibitory circuits by immature ABNs, we hypothesized that the DG neurons of irradiated mice would be hyperactive in comparison with the non-irradiated cohorts. To test this, we computed the calcium activity rates of cells that were active

during recording sessions by integrating their calcium signals over each session and normalizing this to the distance travelled (see methods). In contrast to our expectation, we found that irradiated groups, where neurogenesis was ablated, had significantly reduced calcium activity (Fig. 3d). Keeping with this pattern, EE mice, which have increased neurogenesis, had significantly higher activity than their RC counterparts. Reduced calcium activity rates were found both among the most active neurons, as well as in the less-active quartiles (Supplementary Fig. 4b), which is similar to our findings for tuning (Supplementary Fig. 4a). To better determine how the tuning specificity changed in mice with ablated neurogenesis, we mapped the cumulative activity of each cell at each point in the treadmill to create a tuning curve. We then fit a Von Mises function to the tuning curves of every cell, using the parameters of the fitted curve to compare experimental groups (Fig. 3e, S4c). EE mice had significantly higher goodness-of-fit ($R^2$) than RC mice, while both irradiated groups had significantly lower $R^2$ than their non-irradiated counterparts (Fig. 3f). This could be a result of poorly fitted cells having flatter and/or noisier tuning, resulting in lower information content in irradiated animals[49]. The peak width of fitted curves was not significantly different across experimental groups (Fig. 3g), indicating that differences in spatial information were not due to increased lap-to-lap variance in the location to which individual DG neurons are responsive. However, the peak amplitude of fitted cells was significantly lower in the irradiated cohorts (Fig. 3h), reflecting a dampened place-specific response of individual neurons. These results suggest that the reduction in spatial information content (Figs. 1f, 2e) is driven by two components at the single cell level: an overall reduction in tuning selectivity explained by a decreased number in well-tuned cells, and a reduction in the

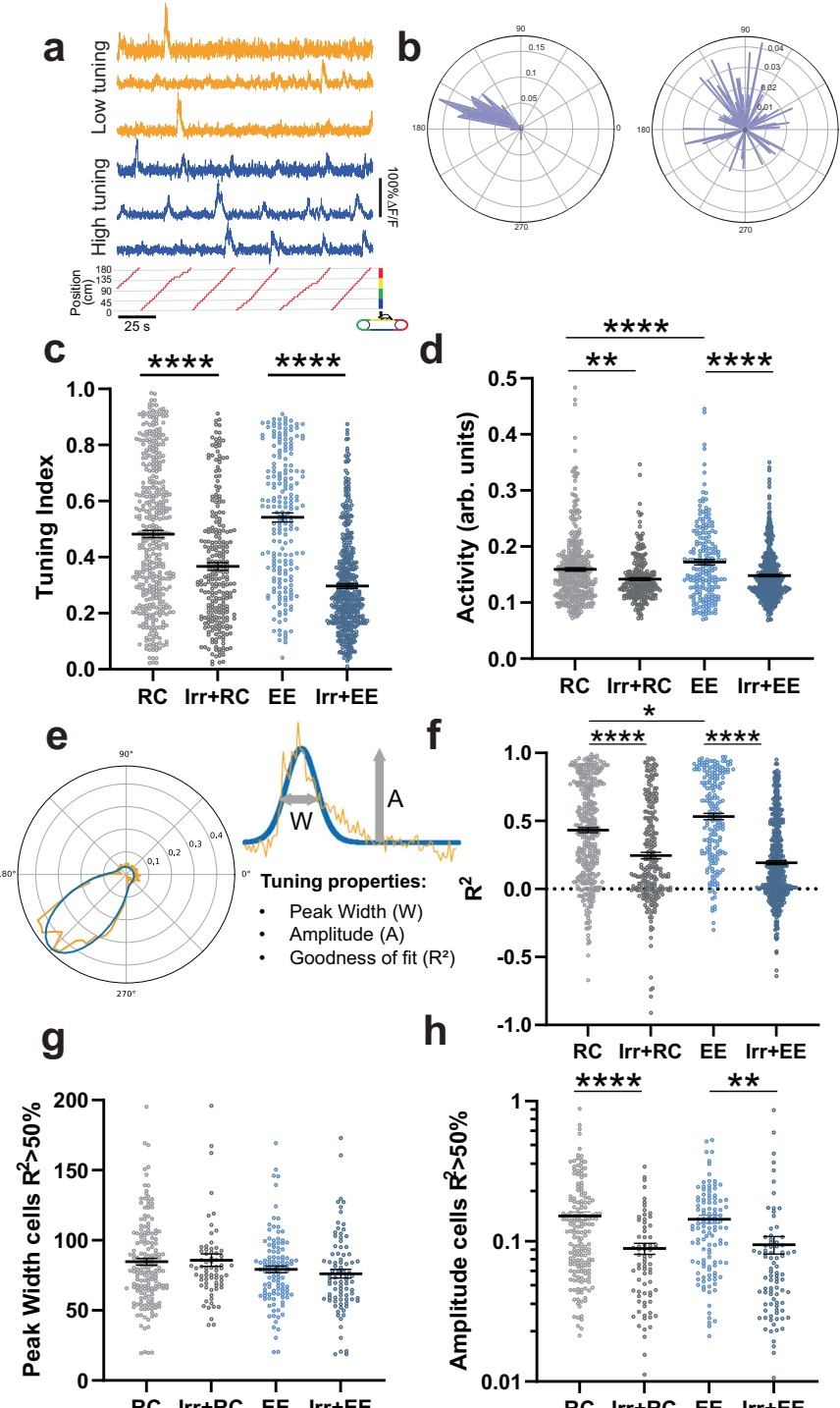

**Fig. 3 | Ablating adult neurogenesis reduces tuning specificity and activity.**
**a** Example calcium traces of cells with low (yellow) and high (blue) tuning indices.
**b** Tuning vectors from cells in A plotted in polar coordinates. **c** Tuning indices of cells in non-irradiated and irradiated groups (RC vs Irr+RC: ****$p < 1/100000$, $n_{RC} = 5$ mice, 277 neurons, $n_{Irr+RC} = 4$ mice, 253 neurons, EE vs Irr+EE: ****$p < 1/100000$, $n_{EE} = 5$ mice, 201 neurons, $n_{Irr+EE} = 4$ mice, 412 neurons). **d** Activity measured as integrated calcium traces normalized to distance travelled (RC vs EE: ****$p < 1/100000$, $n_{RC} = 5$ mice, 277 neurons, $n_{EE} = 5$ mice, 201 neurons, RC vs Irr+RC: **$p < 0.00228$, $n_{RC} = 5$ mice, 277 neurons, $n_{Irr+RC} = 4$ mice, 253 neurons, EE vs Irr+EE: ****$p < 1/100000$, $n_{EE} = 5$ mice, 201 neurons, $n_{Irr+EE} = 4$ mice, 412 neurons).
**e** Schematic of tuning curve properties of individual cells fitted with Von Mises

function. **f** Cross-validated goodness of fit ($R^2$) of tuning curves (RC vs EE: *$p = 0.01617$, $n_{RC} = 5$ mice, 277 neurons, $n_{EE} = 5$ mice, 201 neurons, RC vs Irr+RC: ****$p < 1/100000$, $n_{RC} = 5$ mice, 277 neurons, $n_{Irr+RC} = 4$, 253 neurons, EE vs Irr+EE: ****$p < 1/100000$, $n_{EE} = 5$ mice, 201 neurons, $n_{Irr+EE} = 4$ mice, 412 neurons). **g** Peak width of tuning curves of well-fitted cells. (RC vs Irr+RC: $p = 0.516$, $n_{RC} = 5$ mice, 173 neurons, $n_{Irr+RC} = 4$ mice, 70 neurons, EE vs Irr+EE: $p = 1.159$, $n_{EE} = 4$ mice, 113 neurons, $n_{Irr+EE} = 4$ mice, 85 neurons). **h** Peak amplitude of well-fitted cells (RC vs Irr+RC: $p < 1/100000$, $n_{RC} = 5$ mice, 173 neurons, $n_{Irr+RC} = 4$ mice, 70 neurons, EE vs Irr+EE: $p = 0.00237$, $n_{EE} = 4$ mice, 113 neurons, $n_{Irr+EE} = 4$ mice, 85 neurons). Statistical analyses: Bootstrap (two-sided) and Bonferroni correction for multiple comparisons. Error bars represent mean ± SEM in all plots.

amplitude of the tuning curve with no change in the lap-to-lap variance of the location of the tuning maximum, as observed by the unchanged peak widths. These results suggest that integrating ABNs may be either directly or indirectly modulating the gain of DG neurons, increasing calcium activity rates in a multiplicative manner, which for well-fitted neurons ($R^2 > 0.5$) reflects as a larger increase in calcium activity at the preferred location of each neuron than at non-preferred locations, thereby increasing tuning curve slope and spatial information. Furthermore, because gain modulation can also influence trial by trial variability[50,51], and variability limits single-neuron spatial information, our finding of increased spatial information indicates that the presence of immature ABNs leads to an increased response gain without increasing variability as much. However, the changes we observed could also be due to a non-specific side effect of irradiation, such as increased inflammation. We therefore investigated whether the same circuit effects were present in mice where ABNs were silenced using a chemogenetic approach.

## Acute chemogenetic silencing of ABNs decreases spatial information

Previous studies suggested that synaptic competition between ABNs and mature neurons plays a role in mediating the effects of adult neurogenesis on DG networks. By disrupting the synaptic connectivity of mature neurons, ABNs force the rewiring of the circuit[52–54]. We asked whether ABN-induced changes in spatial information content required ABN activity or were a product of their integration into the DG. To address this question, we acutely silenced a cohort of ABNs by crossing Ascl1-CreERT2 knock-in mice[55] with a line expressing the inhibitory DREADD hM4Di[56,57] in a Cre-dependent manner, allowing us to specifically target a cohort of newborn ABNs. Animals were injected daily with Tam for 3 days at 8 weeks of age (Fig. 4a, b) and housed in EE conditions. Littermates lacking hM4Di expression were used as controls for the unspecific effects of CNO. To assess the efficacy of ABN silencing, and since hM4Di can inhibit synaptic release independently from its effect on action potential firing[56], we tested the effects of CNO-induced silencing on mouse behavior. We used a context discrimination task where the animals were conditioned to associate a specific context with a foot shock (Fig. 4c). In line with previous studies[10,11,29], silencing ABNs resulted in impaired discrimination between a Context A where the animals were shocked, and a novel Context B (Fig. 4d, e). All mice were injected with an AAV encoding jRGECO1a and implanted with an imaging window over the hippocampus 2 weeks after the end of Tam treatment. As in our previous experiments, we found that this approach labeled only mature (> 6 weeks post-mitosis) ABNs, since hM4Di-expressing ABNs did not express jRGECO1a (Supplementary Fig. 5). The bulk of the silenced ABNs was therefore 4–6 weeks post-mitosis, a timepoint when they were found to have a significant impact on behavior[15,16]. Mice were imaged 3.5 weeks after implantation, both in baseline conditions and 30 min after an i.p. CNO injection (5 mg/kg). The exact same field of view was imaged in both sessions. Consistent with our previous results in irradiation experiments, single-cell Fisher information (Fig. 4f), Ca$^{2+}$ activity (Fig. 4h) and tuning index (Fig. 4j) were significantly reduced after silencing ABNs with CNO. Both Fisher information and tuning index were unchanged in control animals that did not express hM4Di (Fig. 4g, k), although Ca$^{2+}$ activity showed a significant increase after injection of CNO (Fig. 4i). This is the opposite direction of change from hM4Di expressing animals and could be due to unspecific effects of CNO. We also fit the tuning curves of DG neurons before and after CNO administration to a von Mises function (Supplementary Fig. 6a, b). The goodness-of-fit ($R^2$) showed a trend for decrease as seen in the irradiated cohorts but no significant difference was found (Fig. 4l, m). However, when only well-fit cells ($R^2 > 0.5$) were considered, the silencing of ABNs resulted in lower goodness-of-fit, while control animals remained unchanged (Fig. 4n, o), which is in accordance with the irradiation experiments. Similar to our previous

experiments, no significant change was seen in the width of the fitted tuning curves (Fig. 4p, q), while their amplitude was decreased upon CNO administration (Fig. 4r, s). Neither parameter was changed in control animals that did not express hM4Di. Given these findings, we conclude that the effect of ABNs on the neural representations of space in the DG is dependent on their activity. Using a linear decoder, we decoded population activity before and after CNO administration. Despite a trend for lower decoding performance after CNO, this difference was not significant (Supplementary Fig. 6c), although this could be due to the fact that some of the mice did not meet the subsampling threshold due to few cells being active in the field of view (Fig. 1f, Supplementary Fig. 3 f). However, population Fisher information, another measure of information at population level (Fig. 2a–d), did show a significant decrease after CNO injection (Supplementary Fig. 6d). Silencing ABNs replicated the main results from the ablation experiments: it decreased Fisher information, Ca$^{2+}$ activity and tuning, while decreasing the goodness-of-fit of tuning curves to a Von Mises function, and reducing their amplitude, suggesting that spatial information changes in the irradiated cohorts were not due to the side effects of irradiation, such as increased inflammation. Taken together, these results indicate that ABNs increase spatial information in the DG by increasing the spatial selectivity of otherwise untuned DG granule cells, and leading to an increase in the gain of DG granule neurons, either directly or through other circuit mechanisms.

## Discussion

Adult neurogenesis is a unique type of brain plasticity that involves the addition of new neurons in response to environmental factors and other stimuli. In the years since ABNs were first reported in the rodent DG[58], several studies have confirmed their presence in humans[59–61] and other species, characterized their cellular and physiological properties, and identified their different contributions to memory tasks. Despite this progress, the processes mediating the contribution of ABNs to behavior remain poorly understood, partially due to the technical difficulties of recording population activity in DG circuits in vivo. More recently, several studies made use of in vivo calcium imaging to determine how the DG encodes space, finding that a portion of DG neurons are spatially selective, although immature ABNs were less selective than their mature counterparts[29,62–64]. In this work we used a similar in vivo imaging approach to determine how the DG spatial code changes when adult neurogenesis is upregulated with EE or, conversely, when neurogenesis is ablated, or ABNs are silenced. We imaged naïve mice during their first exposure to a moving treadmill. Our results showed that 2 weeks of EE housing prior to imaging increases spatial information content in the DG, and that this increase requires ABNs (Fig. 1). This change in neural representations following EE is a new phenotype that potentially explains why EE and similar environmental manipulations are beneficial to memory and learning. We also found that ablating ABNs decreases spatial information content in the DG (Figs. 1f, 2e). We then showed that these ABN-driven changes are mediated by an increase in spatially selective activity at the single cell level (Fig. 3c, d, h). Furthermore, we described a potential dual mode of action for ABNs: increasing both overall tuning selectivity and response gain of DG neurons, which in well-tuned cells results in an increase in activity specifically at the tuning maximum. Finally, we showed that ABN activity is required to elicit the changes in information content in the DG, as both tuning selectivity and spatial information were reduced after acutely silencing a cohort of immature ABNs (Figs. 4f, j). Silencing ABNs also resulted in a decrease in overall Ca$^{2+}$ activity and, in well-tuned cells, a decrease in activity at the maximum of the tuning curve. Overall, these findings suggest that ABNs may contribute to cognition by modulating information encoding in the DG.

ABNs are thought to have a net inhibitory effect on DG activity[30–32], which is regarded as beneficial for tasks such as contextual

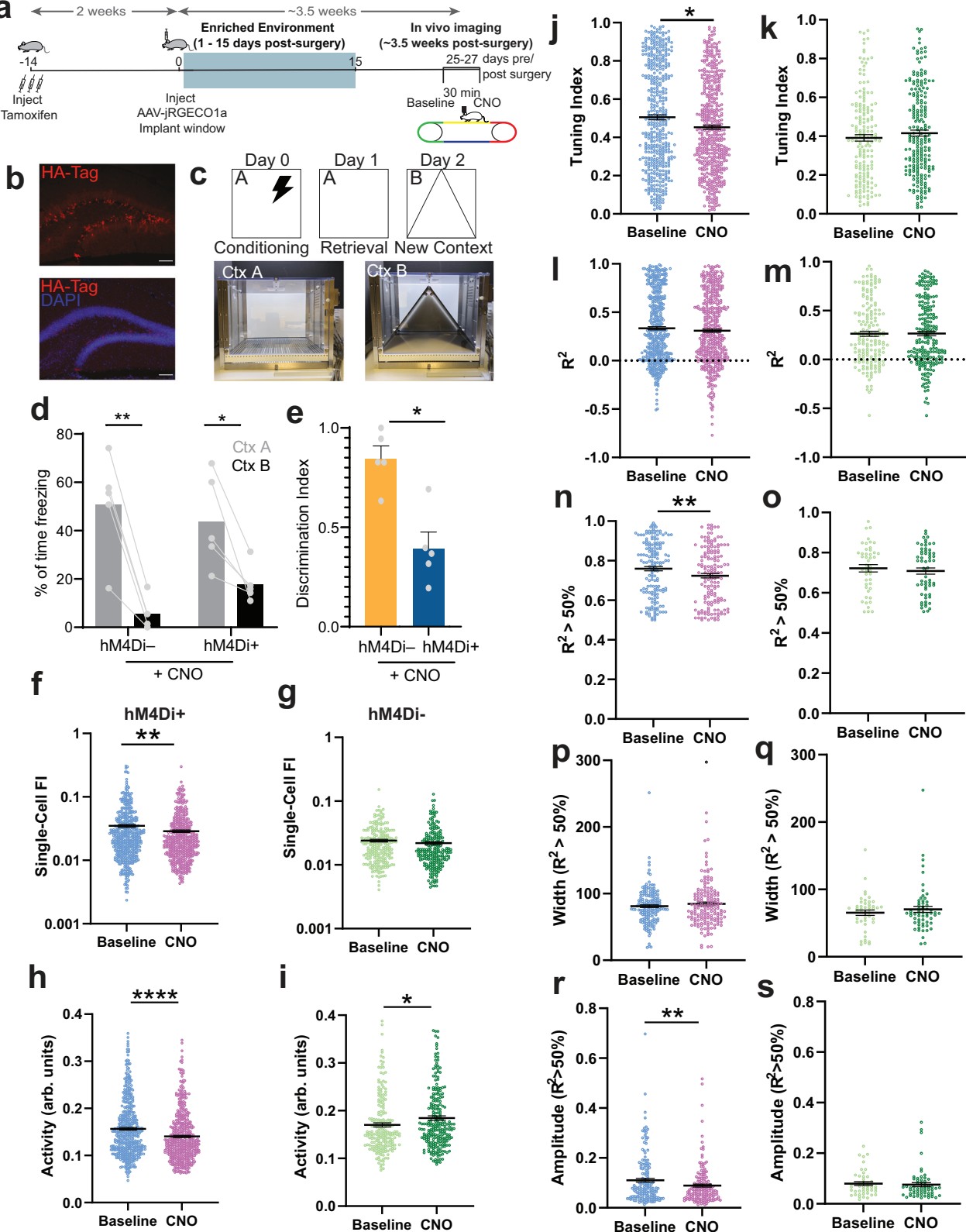

memory discrimination since the resulting increase in sparsity reduces the overlap between neuronal populations responding to different stimuli. The mechanisms behind this inhibitory action are not entirely understood, as immature ABNs are only poorly coupled to feedback inhibitory circuits in the DG[65,66]. Our results stand in contrast with mechanistic models that propose an inhibitory effect of ABNs: we found that increasing the number of ABNs modulates the gain of DG

neurons, leading to higher single-cell calcium activity rates (Fig. 3d). In addition, this increase in activity was spatially selective: it conveyed spatial tuning and increased spatial information by increasing activity near the peak of the tuning curve more than at other locations. Our results do not necessarily mean that the net effect of ABNs in the DG is excitatory, since our analyses only considered neurons that were active during each imaging session while other studies found

**Fig. 4 | Acute chemogenetic silencing of ABNs decreases spatial information content in the DG. a** Experimental timeline. **b** Immunofluorescence images of HA-tag positive neurons (red) and DAPI labelled nuclei (blue). **c** Schematic of contextual fear conditioning (CFC) task and images of context A and context B. **d** Percentage of time spent freezing in shocked and novel contexts (hM4Di- Ctx A vs Ctx B: **$p = 0.0015$, $n_{Ctx A} = 5$ mice, $n_{Ctx B} = 5$ mice, hM4Di+ Ctx A vs Ctx B, *$p = 0.0484$, $n_{Ctx A} = 5$ mice, $n_{Ctx B} = 5$ mice,). **e** Discrimination index of freezing between contexts (hM4Di- (control) vs hM4Di +: *$p = 0.0159$, $n_{hM4Di-} = 5$ mice, $n_{hM4Di+} = 5$ mice). Statistical analyses c) and d): Two-sided Mann-Whitney test, no multiple comparisons correction. **f, g** Fisher information (Baseline vs CNO: **$p = 0.00345$, Baseline vs CNO (hM4Di- control): $p = 0.08291$,). **h, i** Activity (Baseline vs CNO: ****$p = 0.00005$, Baseline vs CNO (hM4Di- control): *$p = 0.01964$). **j, k** Tuning index (Baseline vs CNO: *$p = 0.0173$, Baseline vs CNO (hM4Di- control):

$p = 0.13226$). **l, m** Goodness-of-fit ($R^2$) of the Von Mises function to the tuning curves of all cells (Baseline vs CNO: $p = 0.44973$, Baseline vs CNO (hM4Di- control): $p = 0.47035$). Sample size **f)**, **h)**, **j)** and **l)**: $n_{Baseline} = 9$ mice, 471 neurons, $n_{CNO} = 9$ mice, 461 neurons. Sample size **g)**, **i)**, **k)** and **m)**: $n_{Baseline} = 5$ mice, 228 neurons, $n_{CNO} = 5$ mice, 217 neurons. **n, o** Goodness-of-fit of well-fitted ($R^2 > 0.5$) cells (Baseline vs CNO: **$p = 0.00776$, Baseline vs CNO (hM4Di- control): $p = 0.47035$). **p, q** Peak width of well-fitted cells (Baseline vs CNO: $p = 0.49976$, Baseline vs CNO (hM4Di- control): $p = 0.17618$). **r, s** Peak amplitude of well-fitted cells (Baseline vs CNO: **$p = 0.00636$, Baseline vs CNO (hM4Di- control): $p = 0.3244$). Sample size **n)**, **p)** and **r)**: $n_{Baseline} = 9$ mice, 168 neurons, $n_{CNO} = 9$ mice, 161 neurons. Sample size **o)**, **q)** and **s)**: $n_{Baseline} = 5$ mice, 45 neurons, $n_{CNO} = 5$ mice, 60 neurons. Statistical analyses **f)** thru **s)**: Two-sided bootstrap. Error bars represent mean ± SEM in all plots. All scale bars = 50 μm.

differences in the fraction of active neurons. It is possible that ABNs act to reduce the fraction of active DG neurons while at the same time increasing the firing rate of those same active neurons. Of note, a recent calcium imaging study found that voluntary exercise, which also increases adult neurogenesis, leads to increased calcium activity and spatial information in CA1[67], which is similar to our findings on the effect of EE in DG. Conversely, another study by McHugh et al., using single unit electrophysiology recordings with opto-tagged ABNs, found that optogenetically silencing immature ABNs resulted in increased firing rates of mature DG neurons, as well as principal neurons in CA3 and CA1, with more pronounced effects in the latter two areas[28]. Additionally, the same study found that single-neuron spatial information in all three areas remains unchanged upon silencing of ABNs. Instead, the authors found that silencing ABNs reduces population sparsity throughout hippocampal subfields. These findings appear to be in disagreement with ours, although this could be due to methodological differences: electrophysiology vs. $Ca^{2+}$ imaging recordings in our work, different genetic drivers targeting newborn ABNs (Nestin-Cre vs. Ascl1-Cre-ERT2), different measures of information content (mutual information vs. Fisher information) and our use of EE. Although electrophysiological recordings directly record action potentials with much higher temporal resolution than in vivo $Ca^{2+}$ imaging, the sparse nature of DG activity makes these recordings particularly challenging, resulting in relatively fewer single-unit recordings in DG, which may be reflected in the fact that the strongest effects of silencing ABNs were found in CA3 and CA1. In another study using in vivo $Ca^{2+}$ imaging, Tuncdemir et al. found that ablating or silencing ABNs does not alter population-level fraction of active cells (i.e. sparsity), mean firing rates or spatial information/tuning properties during the formation of cognitive maps[64], which differs from our work and partially differs from the data of McHugh et al. Again, differences in the experimental and analysis methods could account for the different findings: a different implantation surgery that requires the partial removal of CA1, a rewarded treadmill task, and different methods to estimate $Ca^{2+}$ activity and spatial information. In an attempt to reconcile the findings of different labs, we have made the datasets and analysis software used in this publication openly available.

One concern about our AAV labeling approach is that it could potentially target both mature DG neurons and immature ABNs, which raises the question of whether the increase in single-cell calcium activity with EE could potentially be due to more immature ABNs being present in the field of view, their higher excitability resulting in increased calcium activity. However, we verified that immature ABNs were not labeled with jRGECO1a (Supplementary Figs. 2, 5), which is potentially due to their sensitivity to AAV[38]. Furthermore, immature ABNs comprise a very small proportion[6] (<1%) of the total DG granule cell population, and they are less spatially tuned[28,29], whereas we observed a significant increase in tuning. In order to further verify this, we replicated our main results using a labeling approach that targets mature DG neurons specifically (Supplementary Fig. 7) resulting in

calcium sensor expression in neurons that are >6 weeks post-mitosis at the time of imaging. Immature ( < 6 weeks-old) ABNs were not labeled since their axons had either not reached CA3 at the time of the viral injection, or were not mature enough for retrograde transport AAVs, which we confirmed by showing that GFP-labeled ABNs <17 days post-mitosis (i.e. <6-weeks at the time of imaging) were not co-labeled by a retrograde AAV injection into CA3 (Supplementary Fig. 8). This confirms that the changes in neuronal activity leading to improved spatial coding occur in the mature granule cells, and not only in the immature ABNs.

The positive correlation between neurogenesis and DG neuronal activity suggests that the contribution of ABNs to the DG neural code is not solely mediated by feedback inhibitory circuits. In addition to making weak connections to local interneurons, immature ABNs also form synapses with mossy cells, a population of excitatory neurons whose axonal arbors innervate large portions of the DG granule cell layer[68]. It is possible that recurrent excitation through mossy cells may mediate the increase in tuning of DG neurons in mice with elevated adult neurogenesis. Our finding that ABNs act by modulating the gain of DG granule cells offers an additional clue about potential mechanisms. Gain modulation has been shown recently to influence response variability in visual cortex[50] producing diverse effects depending on the stimulus and other factors[69]. In our analysis, the gain increase induced by ABNs did not increase lap-by-lap variability of responses disproportionately, because if it did, then spatial information would decrease despite the higher response amplitude and tuning slope. ABNs may thus act by recruiting a network mechanism that simultaneously increases DG activity and stabilizes it by modulating the balance of recurrent excitation and inhibition[70–73].

In one set of experiments we used irradiation as a method to ablate neurogenesis, which can result in increased inflammation and other side effects. We therefore employed a well-established method of focal irradiation[35], and allowed a one-month period of recovery prior to imaging sessions to allow for inflammation to subside. In order to quantify inflammation we traced DG microglia morphology in tissue collected from EE and Irr.+EE mice that underwent in vivo $Ca^{2+}$ imaging. Microglia are known to undergo morphological changes in response to inflammation, with their processes becoming less arborized[74,75]. Irr.+EE mice had less complex branching, with fewer Sholl intersections, and shorter branches (Supplementary Fig. 9), which is indicative of increased inflammation when compared with EE controls. This increase in inflammation could account for the differences in activity and spatial information in the irradiated cohorts. Although the consequences of inflammation on information encoding are not completely understood, this is a caveat in our data that is mitigated by our ABN silencing experiments. Using a chemogenetic approach to acutely silence ABNs, we obtained similar results as in irradiated animals (Fig. 4), namely a decrease in single-cell activity and single-cell spatial information, as well as a decrease in goodness-of-fit and tuning curve amplitude in spatially tuned cells. However, the effects of silencing were less pronounced than those of irradiation, likely because fewer

neurons are targeted when using chemogenetics and, in addition, neurons that express hM4Di will likely not be completely silenced by CNO administration. Possibly because of this, silencing ABNs did not result in a significant decrease in the performance of our linear decoder (Supplementary Fig. 6c).

Our silencing approach used an inducible transgenic mouse line to express hM4Di in Ascl1+ progenitors that give origin to both DG and olfactory bulb (OB) ABNs. Silencing olfactory bulb ABNs could potentially result in decreased olfactory discrimination, which could in turn impact spatial tuning in the DG as the animals may also use olfaction to navigate the treadmill. However, our irradiation experiments showed a significant difference in spatial tuning when DG neurogenesis is selectively ablated and sub-ventricular zone neurogenesis is spared (Supplementary Fig. 1c), therefore we do not expect that OB ABNs contribute to DG spatial representations. Overall, our results are consistent across the two techniques used to induce a loss-of-function of immature ABNs.

Another potential caveat in our experiments is the use of AAV vectors to deliver the jRGECO1a calcium sensor. Although AAVs have been widely used in a variety of neuroscience applications, including in-vivo calcium imaging, recent findings have demonstrated that intermediate progenitor cells and <2-week-old ABNs can undergo apoptosis when targeted by AAVs[38]. To overcome this caveat, we optimized our protocol to limit the time between AAV injections and imaging. Nevertheless, we performed an additional control experiment using a retrograde AAV vector injected into hippocampal area CA3. The retrograde viral vectors are taken up by the axonal terminals of DG granule neurons, resulting in the expression of jRGECO1a in the ipsilateral granule layer while preserving similar numbers of ABNs in the injected and contralateral hemispheres[38]. Using this approach, we were able to confirm that housing mice in EE elicited an increase in calcium activity and single-cell spatial information content in the DG (Supplementary Fig. 7). Although tuning trended higher in EE mice (Supplementary Fig. 7d), this difference was not significant with our sample size since the retrograde labeling approach resulted in a significantly smaller number of active neurons per field of view. This was likely due to lower labeling efficiency, resulting in lower copy number of the AAV vector and, consequently, lower expression.

In our analysis of the factors that contribute to spatial encoding, we found that noise correlations between neurons had no effect on spatial information content or were even detrimental to it (Fig. 2a–d), although spatial information in the shuffled datasets was not significantly different. This suggests that noise correlations might limit the amount of spatial information encoded by the DG, similar to recent results in CA1[48], although one caveat is that we imaged a relatively low number of active cells when compared with the entirety of the DG coding space, as DG activity is very sparse. Our result is different from, but not incompatible with, a related finding in DG[47], which showed that knowing the structure of noise correlations helps to correctly decode the available information[43,45]. Furthermore, in our data the effects of noise correlation do not appear to be qualitatively modulated by adult neurogenesis, namely, the action of ABNs did not induce a switch from detrimental to beneficial noise correlations.

While our work focused on spatial information, several types of neuronal coding modalities likely coexist in the DG. Neurons in the dorsal DG encode not only position, but also direction of motion, speed[29,47], as well as other sensory cues[62,63], and a population of stress-responsive cells has also been found in the ventral DG[30]. It is possible that ABNs have different effects on these cell populations. Cue-cells[63], which encode sensory features, have been found to be strongly modulated by ABNs during contextual changes[64]. Recordings during other specific behavioral tasks or in the ventral portion of the DG may still reveal different functional roles for ABNs. Alternatively, the gain amplification effect we identified could generalize beyond spatial information and improve encoding across other modalities.

In summary, our results indicate that adult neurogenesis leads to increased response gain in the DG, improving the ability of the granule cells to tune to spatial features and therefore improving spatial information encoding. Our findings demonstrate that the spatial information content in the DG can be modulated even by brief environmental manipulations, such as EE housing, that result in changes in the number of ABNs. The increased spatial information and resulting improvement in the accuracy of neural representations of space in mice with elevated neurogenesis provide a mechanistic circuit-level explanation for their improved performance at many spatial memory tasks[8]. Furthermore, our findings provide a different mechanistic model of how ABNs may contribute to memory, paving the way for further research on adult neurogenesis as a therapeutic target for memory disorders.

## Methods

### Animals
We used C57BL6/J (Jackson Labs Stock #664) mice or, for chemogenetic silencing experiments, the offspring of hM4Di-Dreadd (Jackson Labs #26219) and Ascl1-Cre-ERT2 mice (Jackson Labs #12882). The sex of individual mice is indicated in Supplementary Table 1. All mice were kept on a 12 h light/dark cycle under standard environmental conditions (18–23 °C, 40–60% relative humidity) and were allowed standard chow and water ad libitum. Animals were housed in groups of 3–5 and littermates were divided between experimental groups. Mice assigned to enriched environment (EE) housing were housed in groups of 5–10 in a large 121 × 61 cm enriched cage, containing a feeder, water dispenser, several running wheels, as well as plastic tubes, domes and other structures. Mice assigned to EE experimental groups were housed in EE cages starting the day after the implantation surgery and for a period of 2 weeks, after which they were returned to RC conditions for the remainder of the experiment. Female and male mice were never mixed in the same cage, and all males were housed with littermates. Regular cage (RC) controls were housed in groups of up to 5 mice in standard mouse cages (dimensions 28 cm × 18 cm) containing a wire feeder and a water bottle. Experiments were carried out during the light phase of the cycle. All mice were euthanized with a lethal dose of ketamine followed by transcardiac perfusion for tissue preservation. All procedures were done in accordance with a protocol approved by the Institutional Animal Care and Use Committee (Protocol #: 00001197).

### Focal Irradiation
We permanently ablated adult neurogenesis in the dentate gyrus (DG) by bilateral focal irradiation of the hippocampus using opposed lateral fields (Supplementary Fig. 2) in C57BL/6 J mice (Jackson Labs #664). First, we traced the location of both hippocampi from a thin slice 9.4 T, T1 weighted MRI. Irradiation was performed using a small animal radiation device (SARRP, Xstrahl). To ensure a reproducible treatment setup, the mice were briefly anesthetized with isoflurane and immobilized using a custom fixation system prior to radiation delivery. Cone-beam computed tomography was used to setup the irradiation fields and for calculating individual irradiation times, ensuring accurate and reproducible delivery of the intended irradiation. Hippocampal focusing irradiation was delivered using a 3 mm × 10 mm irradiation field positioned to cover the dorsal part of the brain, avoiding the olfactory bulb and subventricular zone (SVZ). A single dose of 10 Gy (at 2.5 Gy/min) was administered to trigger cell death within the hippocampus[35]. Irradiated mice were allowed one month to recover prior to imaging.

### Viral labeling and window implantation surgery
The right hemisphere dentate gyrus (DG) was labeled with an AAV vector that expressed the jRGECO1a genetically encoded calcium sensor[36] under the control of the CaMKIIα promoter (DJ serotype AAV-

CaMKIIα.jRGECO1, University of North Carolina Vector Core, plasmid kindly donated by Dr. Fred Gage). Mice were anesthetized with isoflurane (induction: 5%, maintenance: 2% in $O_2$ vol/vol, via nose cone) and placed in a stereotaxic frame. The right DG was stereotactically targeted[76] with a pulled-glass micropipette and 950 nl of viral solution ($5.4 \times 10^{12}$ viral particles/mL) were injected with a microinjector (Drummond Nanoject III). This approach labels mature DG neurons but no immature ABNs (Supplementary Figs. 2, 5), presumably due to the toxic effects of AAVs and potential detrimental effects of calcium-sensor overexpression in immature ABNs. AAV vectors can be toxic to transient amplifying progenitors and immature neurons in a dose-dependent manner[38], resulting in a localized permanent depletion of ABNs in areas of high expression at the epicenter of injection, while neurogenesis persists in adjacent dorsal DG areas under the imaging window (Supplementary Fig. 2). We were unable to label ABNs using a genetic approach by crossing Ascl1-CreERT2 transgenic mice[55] with mouse lines expressing GCaMP6f/s in a Cre-dependent manner (Ai95, Ai96[77]), which confirms previous reports from the Sakaguchi Lab[78,79] who were successful only when using the earlier version GCaMP3 which has reduced brightness and signal-to-noise ratio[80]. Similarly, labeling with a Moloney Mouse Leukemia Virus (MMLV) retroviral vector encoding GCaMP6m or jRGECO1a failed to yield any fluorescently labeled ABNs. We therefore speculate that genetically-encoded calcium sensors could also have a detrimental effect on immature ABNs, perhaps due to excessive calcium buffering. In order to control for the toxic effects of AAVs on ABNs we used a retrograde AAV labeling approach injecting AAV-Syn-jRGECO1a (Addgene #100854-AAVrg) into CA3[38]. This method preserves adult neurogenesis since neuronal progenitors and immature ABNs are yet to extend axons to CA3. In order to determine the maturity stage of ABNs labeled with retrograde AAVs, we double labeled newborn ABNs by injecting RV-GFP (MMLV-based vector, Salk Institute Viral Core) into the dorsal DG[76], followed either 17 or 25 days later by an injection of AAV-Syn-jRGECO1a into CA3.

Mice that were imaged in vivo also underwent surgery to implant an imaging 'window': a 3 mm craniotomy was drilled around the viral injection site and a custom-made titanium ring with a glass bottom was placed immediately above the dorsal surface of the hippocampus and anchored to the skull with dental cement[37,81]. The alveus and all hippocampal structures were left untouched during this procedure. A small titanium bar ($9.5 \times 3.1 \times 1.3$ mm) was also attached to the skull in order to attach the animal to the microscope stage. Mice were given carprofen (5 mg/kg) for inflammation and analgesic relief.

### In vivo 2-photon calcium imaging

In vivo calcium imaging was performed 3 to 4 weeks after surgery when mice were 11.5–13.5 weeks of age. We used a two-photon microscope (Thorlabs Bergamo) equipped with a 16 × 0.8 NA objective (Nikon) and a Fidelity-2 1070 nm laser (Coherent) as a light source. Data acquisition was done with ThorImage 4.0 and ThorSync 4.0 software (Thorlabs). Mice were head-fixed and placed on a treadmill belt. For optimum light transmission, the angle of the mouse's head was adjusted to ensure that the imaging window was perpendicular to the optical axis of the objective. Movies of calcium activity were acquired at 15 frames/s using an average laser power of ~120–180 mW, as measured in front of the objective.

We acquired 9 min of calcium activity recordings for each mouse. The animals were imaged as they walked head-fixed on a previously described treadmill[82] which was manually rotated at a speed of approximately $421 \pm 23$ cm/min (Supplementary Fig. 1b). The treadmill consisted of a belt with 4 different textures each 45 cm length (velvet, smooth, 2.5 cm-diameter sandpaper disks of 100 and 60 grit) wound around two wheels. An optical rotary encoder was attached to the axel of one wheel to measure the movement of the belt, enabling the estimation of the position of the mouse along the belt. Four radio

frequency identification (RFID) tags were attached to the belt at transition zones between textures to correct the accumulating error of position estimation by the rotary encoder data. Data from the treadmill was acquired using digitizer hardware (National Instruments) and the ThorSync software (Thorlabs). The treadmill and microscope setting were completely novel to the mice, which were allowed to sit still or freely move and explore the treadmill during initial setup, but were required to walk during imaging, as the treadmill was rotated. Throughout the imaging session, the mice remained in the dark, in an enclosed box built around the microscope.

### Silencing of adult-born neurons

To study the effects of adult neurogenesis on hippocampal circuits we recorded DG activity (or contextual conditioning behavior) before and after silencing a cohort ABNs. We targeted ABNs by crossing Ascl1-CreERT2 transgenic mice[55] (Jackson Labs #12882) with a line expressing the inhibitory DREADD hM4Di[56,57] in a Cre-dependent manner (Jackson Labs #26219). Mice with the two alleles allowed us to specifically target a cohort of newborn ABNs by injecting Tamoxifen (Tam) i.p. over the course of 3 days to express Cre in Ascl1+ cells, thereby inducing Cre expression. This approach resulted in hM4Di expression in both DG and SVZ newborn ABNs. We acutely silenced ABNs by administering the specific hM4Di ligand clozapine-N-oxide (CNO)[56] 30 min prior to imaging behavioral testing, as described above. CNO was dissolved at a concentration of 1 mg/mL in saline solution (0.9% NaCl) and was injected intraperitoneally at a dose of 5 mg/kg[83]. We used littermates lacking the Cre allele, and therefore hM4Di expression, as controls for the unspecific effects of CNO and Tam.

### Analysis of calcium imaging data

Calcium imaging fluorescence intensity data was extracted from movies using the Suite2p (version 0.9.2) open-source software suite[84]. Briefly, all movies were registered for motion correction, the cell contours of active cells were detected and calcium traces extracted for each cell. Movement (rotary encoder) and texture (RFID) data were matched with the corresponding imaging frames. The position of the mouse on the treadmill was determined by calculating the cumulative sum of the treadmill rotation signal for every frame of the calcium imaging movie.

To decode the spatial position from the neuronal population activity, we used a logistic regression model[45], trained on 75% of the data and tested using the remaining 25%, cross-validating using 10 random splits of the data to account for overfitting. We determined that decoding the position from unfiltered fluorescence traces yielded the best results and dynamic range, as either filtering, thresholding or deconvolution results in consistently very high or very low decoding accuracy (Supplementary Fig. 1c, d, e). We also verified that decoder performance was similar on train and test data, indicating minimal overfitting (Supplementary Fig. 1f, g). The position data was segmented into 20 bins and ΔF/F unfiltered calcium traces were used as input data. The population activity was projected onto the decoder weights to obtain a 1D signal and compute d-prime squared (which is equivalent to linear Fisher information).

Single-cell Fisher Information (FI) was calculated using the ΔF/F unfiltered fluorescence calcium data using previously published methods[45]. Briefly, a bias-corrected signal to noise ratio was computed, where the signal is the square of the difference of the mean activity at two locations on the treadmill, and the noise is the average variance of the activity at each location. The position data was segmented into 20 bins.

Tuning indices were calculated using deconvolved firing rates[85], which were thresholded to 2 σ of the baseline, so that every point not significantly above that noise threshold is set to zero. The treadmill band was segmented into 100 position bins and the putative firing epochs were mapped to these bins according to the location of the

mice on the treadmill in order to generate a tuning vector for each cell[29]. The mean of the thresholded firing rates at each location was calculated for every neuron and normalized to the time the mouse spent at that position. The tuning index was defined as the modulus of this normalized tuning vector.

Calcium activity was calculated on a per-cell basis: ΔF/F data was first filtered with a third order Butterworth lowpass filter by applying the filter to the data both forwards and backwards to compensate for phase shifts. Significant calcium transients were determined as the consecutive frames that start when the ΔF/F fluorescence signal rises 2 standard deviations (σ) above the rolling-mean baseline and end when the signal drops below 0.5 σ. The significant transients were then removed, and the remaining calcium trace was used to calculate a new rolling mean baseline, iterating through this process 3 times. The resulting significant calcium transient was used to determine the activity as the cumulative sum of the trace for each cell. The result was normalized to the total distance traveled by the mouse.

### Curve fitting analysis

To study the properties of tuning of every cell, we first generated a tuning curve by mapping the unfiltered ΔF/F fluorescence data of each cell to one of 20 angular position bins on the treadmill belt. This fluorescence data was averaged over all laps. This circular tuning curve was fitted with Von Mises function:

$$B + Ae^{k(\cos(x-\varphi)-1)} \tag{1}$$

Where B is an offset parameter, A is the amplitude of the peak, $k$ is a measure of concentration and φ is the location of the peak of tuning. The parameters of the function were optimized numerically by minimizing the sum of squared differences between data and model. The goodness of fit was calculated and cross validated as follows:

$$R^2 = 1 - (ss\_res/ss\_tot) \tag{2}$$

Where $ss\_res$ is equal to the sum of the squares of the residuals and $ss\_tot$ equals the sum of the squares of the differences from the mean. Note that this value of $R^2$ is normalized between 0 (null model, i.e. $ss\_res\_null = ss\_tot$) and 1 (oracle model, i.e. $ss\_res\_oracle = 0$): a $R^2$ value of zero corresponds to a model where none of the variance can be explained by the Von Mises function and 1 corresponds to an error equal to zero, predicting all of the data points. The cross-validation was done using 75% of the data in each bin to train and 25% of the data to test the model. The width of the peak was determined as the circular variance using the function below:

$$V = 1 - \frac{I_1(k)}{I_0(k)} \tag{3}$$

Where $I_1$ is the Bessel function of order 1 and $I_0$ is the Bessel function of order 0[86].

### Immunohistochemistry and imaging of fixed tissue

In order to quantify the numbers of ABNs we labeled brain tissue with an antibody against Doublecortin (DCX) (CST 14802 S, 1:800 or Abcam ab18723, 1:500). To quantify hM4Di expression, we stained brain tissue with an antibody against HA-Tag (CST 3724 S, 1:800). To quantify microglia branching, we stained brain tissue with an antibody against Iba1 (Fujifilm Wako 019-19741, 1:500). Briefly, all animals were perfused with ice cold 0.1 M phosphate buffer saline followed by 4% paraformaldehyde (PFA) and were then post-fixed in 4% PFA for 24 h. Brains were then cryoprotected in 30% sucrose and the hippocampus was sectioned at 40 μm thickness on a freezing microtome, three sections were stained for each mouse, all within the implanted region of the hippocampus. Sections were rinsed three times in 0.1 M PBS (pH = 7.4)

and incubated in a blocking solution (10% goat serum, 0.3% Triton-X in 0.1 M PBS) for one hour. The sectioned tissue was then incubated in a primary antibody in block for 48 h, followed by three rinses in 0.1 M PBS. The sections were then incubated for 2 h in Alexa 488 (A11034), Alexa 568 (A11011), or Alexa 633 (A21070) goat anti-rabbit secondary antibodies (Invitrogen) diluted at 1:500 in 0.1 M PBS, rinsed three times in 0.1 M PBS, counterstained with 300 nM DAPI in 0.1 M PBS, rinsed again in 0.1 M PBS, and then mounted and cover-slipped with Fluoromount-G (Southern Biotech). The tissue was then imaged on a Zeiss Axio Imager.A2 fluorescence microscope or a Zeiss LSM 880 Airyscan confocal microscope. Exposure time and excitation light intensity were kept constant for each experiment. Analysis of the branching of Iba1-stained microglia was done with the Imaris Software (Oxford Instruments). The position of microglia soma was estimated using the spot function and branching was traced using the filament reconstruction function, using manual curation to discard cells that branched outside of the field of view, or whose the reconstruction was inaccurate.

### Contextual discrimination task

Mice were placed in a fear conditioning chamber within a sound-attenuating cubicle (Med Associates VideoFreeze). For conditioning, mice were allowed 3 min of free exploration of a pre-cleaned cube-shaped chamber with a grid floor (context A) before receiving three mild foot-shocks (2 s, 0.7 mA) spaced 60 s apart. Mice were returned to their home cage 30 s after the last shock. Contextual fear memory was tested 24 h later by re-exposing mice to context A for 3 min of free exploration (no shocks). Forty-eight hours after conditioning, mice were tested on discrimination of a novel context B (plastic floor, A-frame geometry, and scented). In this session, mice were allowed 3 min of free exploration (no shocks). Context discrimination was measured as a discrimination index, DI: (Time freezing in A – Time freezing in B)/(Time freezing in A + Time freezing in B). In silencing experiments, we expressed hM4Di specifically in a cohort of immature ABNs so they could be silenced by the selective ligand CNO. Conditioning and re-exposure to context A took place without CNO but were injected with CNO 30-40 min prior to exposure to the novel context B. To control for unspecific effects of CNO we used mice that lacked a Cre allele, and therefore did not express hM4Di (hM4Di-).

### Statistics and reproducibility

Non-parametric two-tailed Mann-Whitney, paired t-tests, one-way ANOVA and one-way non-parametric Welch's ANOVA tests were used to compare between mouse groups, as described in each figure legend and as appropriate to the design of each experiment. When pooling cell data from different mice into a single experimental group, significance testing was done using a muti-level two-tailed bootstrapped approach that took data nesting into account, as follows. To assess a significant difference between two experimental conditions (e.g. EE and RC), the null distribution was constructed for each condition as follows: The data from both conditions was combined into one group. The animal and data value were sampled in each condition, a total equivalent to the number of data values within the condition. The mean was then calculated. This process was repeated for each condition and the difference between the null distributions generated for each group was calculated. 100,000 bootstraps were generated. The empirically observed value of the difference between conditions was then compared to the null distribution. The source code of the nested-bootstrap is provided as part of the analysis code used in this publication. The statistical significance level (α) was set at 0.05. Bonferroni's correction for multiple comparisons was applied to all bootstrapped data. One-sample t-tests with Bonferroni's correction for multiple comparisons were used to determine differences between group means and chance performance in contextual fear conditioning experiments. Representative immunofluorescence images (Figs. 1c, 4b)

depict typical results of experiments with several biological replicates (Fig. 1c: $n_{RC}$ = 5 mice, $n_{Irr+RC}$ = 4 mice. Figure 4b: $n$ = 5 mice).

## Reporting summary

Further information on research design is available in the Nature Portfolio Reporting Summary linked to this article.

## Data availability

All the data generated in this study have been deposited in the Zenodo database and links to individual datasets can be accessed at https://github.com/GoncalvesLab/Frechou-et-al.-datasets (https://doi.org/10.5281/zenodo.10985968)[87].

## Code availability

The analysis code used in this study has been deposited on Github and can be accessed at https://github.com/GoncalvesLab/Frechou-et-al-Neurogenesis (https://doi.org/10.5281/zenodo.10986236)[88].

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

## Acknowledgements

We thank Dr. Jake Jordan, Roland Ferger, Elizabeth Wood, Jade Rhoads, Dr. Maria Gulinello, Kevin Fisher, Dr. Mimi Kim and Dr. Sacha Sokoloski for technical advice, assistance and discussions. M.A.F. was supported by a Fulbright Scholarship. M.A.F. and K.D.M. were funded by The Einstein Training Program in Stem Cell Research from the Empire State Stem Cell Fund through New York State Department of Health Contract C34874GG. J.T.G. was supported by the Whitehall Foundation (Research Grant 2019-05-71) and the National Institutes of Health (NINDS R01NS125252). R.C.C. was supported by the National Institutes of Health (NEI R01EY030578, NIDA RF1DA056400). Confocal microscopy experiments were supported by a shared instrumentation grant (1S10OD025295-01A1). This article is dedicated to the memory of Dr. Paul S. Frenette (1965-2021) who provided invaluable encouragement and support for this project.

## Author contributions

Project conceptualization: M.A.F., J. T. G.; Experimental design and data acquisition:, M.A.F., S.S.M., E.A.H. S.G., W.A.T., J.T.G.; Data analysis: M.A.F., R.C.C, E.A.H, J.T.G.; Writing, editing and discussion:, M.A.F., K.D.M., R.C.C., J.T.G.

## Competing interests

The authors declare no competing interests.
