## [Peer Review File · Nature Communications]

Adult neurogenesis improves spatial information encoding in the mouse hippocampusReviewers' comments:

Reviewer #1 (Remarks to the Author):

In this study, Frechou et al., set out to determine the role of adult neurogenesis on the spatial tuning properties of dentate gyrus neurons. To achieve this, they use a variety of methods to modulate neurogenesis and the activity of newly generated neurons. They find that DG GCs from mice housed in enriched environments have increased spatial information compared to mice in regular cages (RC). Furthermore, they find that by focally irradiating or chemogenetically silencing adult born neurons (ABNs), this leads to a decrease in spatial information in DG GCs, and a decrease in calcium activity rates of DG neurons, resulting in decreased amplitude of place specific responses. These results address an ongoing discussion in the field as to how adult generated granule cells impact the overall hippocampal circuit. As presented these results appear to contradict an emerging model in which adult generated neurons reduce activity in the DG and increase the sparseness of spatial representations (for example recent work from the Dupret lab). However, in many instances the experimental data fall short to fully support the model proposed by the authors. These concerns are covered below:

Major Concerns:

-It is this reviewer's opinion that it is necessary to differentiate between adult generated neurons and mature neurons in the recordings. Since ABNs could be labeled with the methods used in this paper, and has been done by others using alternative techniques, this is experimentally feasible. As the aim of the study is to understand what contribution ABNs have on hippocampal place encoding, authors need to directly record from ABNs during the task to assess how these neurons contribute to the decoding performance in the population. This is especially true when considering that a small number of neurons per mouse were recorded and that a fraction of these that have different properties could have an outsized effect. How does one really know the impact on the neural coding in the mature GCs if we don't know how many young neurons are imaged in the EE condition. With only 42 cells subsampled per mouse, if a few of these are abGCs presumably that could make a substantial difference. Use of retro-AAV (as used in S5, with minimal characterization) is not appropriate as this synapses on CA3 neurons are present at 2wks of cell age, and no quantification here is provided to show the age of the infected neurons in that experiment.

- In this study ~1ul of AAV is injected into the DG to study the contribution of ABNs to GC function. However, previous work (Johnston et al. 2021 eLife, an author shared here) argued that AAV transfection (with many payloads, including jRGECO), resulted in complete ablation of neurogenesis. If that publication is correct, would that mean in all imaging experiments there are no ABNs? Thus, it is of profound importance to quantify neurogenesis in their AAV infected mice not only so that the results here can be interpreted, but also to correct/revise the record on the use of AAV in the DG in studies where neurogenesis is studied.
- To image neural correlates of spatial encoding, the authors placed head fixed mice on a treadmill with four different textures (velvet, smooth, 2.5 cm-diameter sandpaper disks of 100 and 60 grit). However, neurons responding at these cue locations may be directly driven by the stimulus irrespective of location (see Tunçdemir 2022). Thus some analysis of cue vs spatial tuning on existing data could address this point.
- Authors show decoding accuracy for the first two figures, however do not show it for the rest of the figures. Authors should show decoding accuracy changes across groups to accompany figs 4 and 5.
- Overall, there are many instances regarding lack of detail when it comes to the methods described. For one, is it not clear what statistical analyses were used for a number of results presented. In addition, there were times when authors elected to do ANOVAs on four groups, and others where they split the groups for separate analyses. Authors should combine all groups into one plot and run appropriate ANOVAs with corrections for multiple comparisons. Additional detail is needed across the

other methods as well.

- There were multiple instances of authors not showing the histological confirmation of sensor expression and window placements across experiments. For example, was the DCX staining in 1C from an imaged mouse? In general, Fig 1-2 needs quantification of DCX in the imaged mice, with representative figures.
- No discussion was provided regarding recent work from McHugh et al., 2022 Nature Neuroscience which shows suppressing abGCs increases dentate gyrus principal firing rates. Authors should address the differences between this study and the previous.

Minor Concerns:

- Figures 1 and 2 should be combined since it seems the RC and EE groups were used in both Fig 1E and 2D.
- Figure S2 is missing scale bar, same animals irradiated or are these a different group of mice?
- Figure 2A/B needs better histology, IRR group looks like it has some background noise that could be masking positive staining.
- Fig 1D is not great representation of data; add compartmentalization of different textures in 1D and add legend to illustrate which traces would be considered place cells.
- Figure 5A → didn't write in when they injected tamoxifen
- Data from the DREADD experiment is underpowered with 2 mice in control group. IN general the numbers are low here, as other experiments are with only 4 mice.
- Across the experiments, Authors describe the effect of single cell spatial tuning changes becoming more tuned with EE, however the authors never show the proportion of neurons that are place cells using traditional metrics, to illustrate that more GCs are spatially tuned, so the reader can compare these results to existing literature..
- Authors point out in page 10 that inflammation from irradiation could be an issue, a quantification of Iba1+ or other markers would help ensure this is not a major issue.

Reviewer #2 (Remarks to the Author):

The manuscript entitled "Adult neurogenesis improves spatial information encoding in the mouse hippocampus" by Frechou and colleagues investigates the role of adult-born dentate granule cells in processing spatial information, a long-standing question with broad implications across disciplines in neuroscience. The authors used a treadmill containing different sensory stimuli to simulate spatial processing while analyzing the activity of dentate granule cells using two-photon calcium imaging in head fixed mice. Mice were exposed to an enriched environment (EE) to increase neurogenesis and compare spatial encoding of dentate granule cells with that of regular caged mice with basal levels of adult neurogenesis, and the content of spatial information was estimated using a linear decoder algorithm. The authors first observed that decoding accuracy of granule cells was higher in mice exposed to EE than controls, suggesting that new neurons would enhance spatial encoding. Then, focal irradiation was used to prevent neurogenesis in the context of an exposure to EE, which this time resulted in no effect probably due to the lack of adult-born neurons. Finally, the involvement of new neurons in enhancing spatial information after EE was interrogated using chemogenetic inhibition of adult-born neurons using the DREADD HM4Di, which again suggested a role for adult neurogenesis. Experiments are overall well conducted and the flow of the manuscript is adequate. However, there are major experimental caveats that will need to be addressed to clarify data and conclusions.

Specific comments:

1) Fig. 2D: please explain/discuss the effect of irradiation on RC mice. Is this due to the elimination of basal neurogenesis? Can you exclude neurogenesis-unrelated effects of irradiation (i.e. inflammation)?

2) In several plots along the manuscript, mean \pm -SEM bars seem out of place. See, for instance, Fig. 3E, where all bars seem too high compared to the scatter density. Please verify and correct all plots.

3) It is very important to estimate and discuss and analyze the data in light of the putative age of the neuronal populations being affected by the experimental manipulations. This is a critical point in the manuscript.

a) Irradiation; mice were irradiated 4 weeks before the exposure to EE, and 7.5 weeks before behavior/imaging experiments. Thus, these mice would entirely lack adult neurogenesis and miss all adult-born neurons up to 7.5 weeks of age.

b) EE exposure: EE is known to enhance survival of immature neurons. What would be the age of these neurons at the time when recordings were done? Ideally, this age should be measured, but it might be inferred if one would consider that EE would act when neurons are around 2-3 weeks old. Populations enhanced by 2 weeks of EE would perhaps be around 5 to 7 weeks old.

c) HM4Di silencing: in Fig. 5, *Ascl1-CreERT2/flox-hM4D* mice were TAM-induced at 8 weeks of age, labeling neurons born at around right this time (0 days old). Mice were exposed to EE simultaneously (presumably enhancing the 2-3 w neuronal population?). Thus, CNO would be silencing neurons between 12 to 27 days, whereas EE would be enhancing the 5-7 weeks population (but not the cohort expressing HM4D).

CNO treatment produced, in general, very limited effects when compared to focal irradiation (see for example, Fig. 5M and compare to 4F). CNO would only be silencing neurons up to 27 days, whereas irradiation would be eliminating all new neurons. Thus, these different experimental designs interrogate neuronal non-overlapping populations of granule cells. To target roughly similar populations and attribute the lack of EE effect in IRR to neurogenesis, chemogenetic silencing should be directed to that same neuronal population (5-7 weeks old). Perhaps under those conditions the effect of CNO on spatial encoding would be more robust and reliable.

Reviewer #3 (Remarks to the Author):

Frechou and colleagues report a study describing how neurogenesis induced by an enriched environment can contribute to the coding of spatial changes in a mouse model. Highly clinically relevant results were obtained using state-of-the-art techniques.

The authors know that a radiation dose of 10 Gy induces inflammation. Their protocol includes a month-long recovery period before imaging sessions, which they claim would significantly decrease

inflammation. To validate this hypothesis, the level of radiation-induced inflammatory markers (such as IL-1 β , IL-6, TNF α) could have been measured in additional groups before irradiation, 6 hours after irradiation and 30 days later. On the other hand, the reviewer agrees that their acute chemogenetic silencing of ABNs supports results obtained after ablating by radiation the neurogenesis in dentate gyrus.

Irradiation of a specific and small area in the mouse brain is difficult. A state-of-the-art animal irradiator was used to delivery a bilateral radiation dose. Consequently, a significant radiation dose only 20% to 30% lower than delivered in the dentate gyrus was deposited in the surrounding brain areas. The level of damage in these could have been assessed by histopathology. A stereotactic irradiation of the radiation dose could have been explored.

Minor comments

- 1) Mention in the captions the length corresponding to the white bar added to the figure, and the meaning of the star "*", **, ***".
- 2) Figure 2: Authors are encouraged to add arrows to indicate where DCX positive cells are located.

We sincerely thank the reviewers for their careful reading of our manuscript and for their thoughtful feedback. In the revised manuscript we have included data from additional experiments and analyses, as suggested by the reviewers. We believe that these additions considerably improved the manuscript and effectively address all of the reviewers' concerns. Below, in blue, we respond to all of the reviewers' individual comments:

Reviewer #1 (Remarks to the Author):

In this study, Frechou et al., set out to determine the role of adult neurogenesis on the spatial tuning properties of dentate gyrus neurons. To achieve this, they use a variety of methods to modulate neurogenesis and the activity of newly generated neurons. They find that DG GCs from mice housed in enriched environments have increased spatial information compared to mice in regular cages (RC). Furthermore, they find that by focally irradiating or chemogenetically silencing adult born neurons (ABNs), this leads to a decrease in spatial information in DG GCs, and a decrease in calcium activity rates of DG neurons, resulting in decreased amplitude of place specific responses. These results address an ongoing discussion in the field as to how adult generated granule cells impact the overall hippocampal circuit. As presented these results appear to contradict an emerging model in which adult generated neurons reduce activity in the DG and increase the sparseness of spatial representations (for example recent work from the Dupret lab). However, in many instances the experimental data fall short to fully support the model proposed by the authors. These concerns are covered below:

Major Concerns:

-It is this reviewer's opinion that it is necessary to differentiate between adult generated neurons and mature neurons in the recordings. Since ABNs could be labeled with the methods used in this paper, and has been done by others using alternative techniques, this is experimentally feasible. As the aim of the study is to understand what contribution ABNs have on hippocampal place encoding, authors need to directly record from ABNs during the task to assess how these neurons contribute to the decoding performance in the population. This is especially true when considering that a small number of neurons per mouse were recorded and that a fraction of these that have different properties could have an outsized effect. How does one really know the impact on the neural coding in the mature GCs if we don't know how many young neurons are imaged in the EE condition. With only 42 cells subsampled per mouse, if a few of these are abGCs presumably that could make a substantial difference.

Based on our current understanding of the properties of ABNs, this is unlikely. Immature ABNs are thought to be less tuned and carry less spatial information than mature neurons, therefore, we would expect that imaging more ABNs would reduce the decoding accuracy. Furthermore, we see an increase in Fisher information in our EE group, an independent measure of spatial information that is based on data from all active neurons and not likely to be biased by a small group of ABNs. However, we recognize that this is a valid point and now provide conclusive evidence that no immature ABNs are labeled with the methods used in this paper, effectively addressing Reviewer 1's concerns (Fig. S2, S5). Both DCX-expressing cells and genetically-labeled hM4Di-expressing neurons, lack jRGECO1a expression, probably because of their sensitivity to AAV vectors (and possibly to calcium sensors – see below) the surviving neurons are those that aren't infected. This is reflected in the fact that the majority of DCX or hM4Di-expressing neurons are located in areas of lower jRGECO1a expression.

It would indeed be ideal to record from ABNs and mature neurons simultaneously, but we and others in the field doubt that this is as easily achievable as the reviewer suggests. While it is true that earlier publications, notably the pioneering work of the Hen, Kheirbek and Losonczy labs (Danielson et al., 2016, cited in our manuscript) claimed to image ABN activity, more recent findings showed that using AAVs to label these neurons at ages < ~2 weeks after-mitosis, results in their death in great numbers (Johnston et al., 2021). This cell death is in addition to the effects of inflammation from the viral injection and appears to be caused by the sensitivity of transient amplifying progenitors to viral ITRs. The sensitivity of transient progenitors and immature ABNs to AAVs is heavily dose- and timing-dependent, which is why we and

others were able to use AAVs in our work. Several labs have independently confirmed these findings. In our experience it is impossible to image ABNs at < 6 weeks post-mitosis by labeling them with AAVs. Our lab has made multiple attempts to label *Ascl1-CreERT2xAi9* (i.e. *tdTomato*-expressing) ABNs with AAVs to express *GCaMP* or *jRGECO* and we never found any instance of clear co-localized signal. Several other groups reported similar results (eg. Kumar et al., 2020, Tuncdemir et al., 2023, and personal communications from other scientists). While ABNs do survive in areas where the expression of the calcium sensor is lower, some areas of high expression close to the epicenter of AAV injection are depleted of ABNs, as we now show (Fig. S2). Of note, in their latest publication (Tuncdemir et al., 2023) the Hen and Losonczy groups do not image newborn neurons and in the histology image provided the 4-week-old immature ABNs do not co-localize with *GCaMP7* signal (see Tuncdemir et al. 2023 Fig. 2B), as was the case in our experiments.

Also of note, we tried to label ABNs without resorting to AAVs by using retroviral vectors (eg. *RV-jRGECO1a*). Disappointingly, we never saw any ABNs expressing calcium sensors when using this method (now mentioned in the methods section since we think this is useful information for the field). In a different attempt, we crossed *Ascl1-CreERT2* mice (which label some Type 1 adult neural stem cells, as well as transiently amplifying progenitors) with the *Ai95/Ai96 flox'd* mouse lines to express *GCaMP6f/GCaMP6s* in ABNs. This approach also yielded no *GCaMP6* labeled neurons. These results are in agreement with the findings of the Sakaguchi Lab in Kumar et al. (2020) where they state "*We tested several versions of GCaMP including GCaMP3, 6s, 6f, 7f and 8, but only GCaMP3 showed consistent expression in young ABNs [...]*".

Given these results, there are concerns in the field is that the strong affinity for Ca^{2+} of recent *GCaMP* sensors leads to intracellular changes in Ca^{2+} buffering which are toxic to immature ABNs (see Carrier-Ruiz et al., 2021). This effect is in addition to the toxic effects of AAVs. Therefore, we now believe that *GCaMP3*, with its lower calcium affinity, is the only genetically-encoded calcium sensor proven to allow recordings from immature ABNs, which is a concern because this older version of *GCaMP* does not reliably detect single action potentials. In fact the fraction of active neurons detected with *GCaMP6s* in cortical recordings is 5-fold higher than with *GCaMP3* (see Chen et al., 2013 Fig. 2e). More importantly, *GCaMP3* is much less bright than newer *GCaMPs* and has a dramatically lower signal-to-noise ratio. This makes it incompatible with our approach of imaging the DG through the intact CA1 (Pilz et al., 2016).

Use of retro-AAV (as used in S5, with minimal characterization) is not appropriate as this synapses on CA3 neurons are present at 2wks of cell age, and no quantification here is provided to show the age of the infected neurons in that experiment.

We have addressed this point with new data in our revised manuscript (Fig. S8). While it is true that the axons of ABNs are present in CA3 at 2 weeks of age, they will not immediately take up retroAAV, which we found out after many attempts to label them. We now provide evidence that ABNs < 6 weeks of age (labeled using *RV-GFP*) are not infected by retroAAVs in our experiments. We show that the timing of the retroAAV infection is crucial, as *RV-GFP*-labeled neurons are not infected if the retroAAV injection is done 17 days post-mitosis (6 weeks at imaging), but will be infected when the retroAAV injection is done 25 days post-mitosis (7 weeks at imaging, Fig. S8). Therefore we can confidently state that no new neurons were included in the retroAAV experiment.

- In this study ~1ul of AAV is injected into the DG to study the contribution of ABNs to GC function. However, previous work (Johnston et al. 2021 eLife, an author shared here) argued that AAV transfection (with many payloads, including *jRGECO*), resulted in complete ablation of neurogenesis. If that publication is correct, would that mean in all imaging experiments there are no ABNs? Thus, it is of profound importance to quantify neurogenesis in their AAV infected mice not only so that the results here can be interpreted, but also to correct/revise the record on the use of AAV in the DG in studies where neurogenesis is studied.

The reviewer makes a good point and we now provide more details about our labeling strategy in order not to sow confusion in the field. Our results are in agreement with the previous study detailing how the

effect of AAVs (or their ITRs) on adult neurogenesis **depends on viral titer and the timing of viral injection (Johnston et al., 2021)**. There is reduced adult neurogenesis and death of immature ABNs at the very epicenter of the injection, whereas neurogenesis continues in immediately adjacent areas (still under implant) where virus expression is lower (Fig. S2). We have quantified neurogenesis in slices that include the imaging implant (shown in lower magnification image Fig S2A), and that are labeled with jRGECO1a. Our labeling methods are similar to those used in Tuncdemir et al. 2023 (Fig. 2B in that paper).

- To image neural correlates of spatial encoding, the authors placed head fixed mice on a treadmill with four different textures (velvet, smooth, 2.5 cm-diameter sandpaper disks of 100 and 60 grit). However, neurons responding at these cue locations may be directly driven by the stimulus irrespective of location (see Tuncdemir 2022). Thus some analysis of cue vs spatial tuning on existing data could address this point.

We now make a mention of the fact that some of the neurons we image may be cue neurons as described by Tuncdemir et al. (2022). However, it is impossible to differentiate cue cells without further experiments because we would need to omit a cue from some of the runs and/or introduce a cue at different places along the track. Also, it bears keeping in mind that cue cells are usually not differentiated in other studies of hippocampal place encoding, partly because it is impossible to fully differentiate them without varying cues independently of the tactile input from the treadmill belt.

- Authors show decoding accuracy for the first two figures, however do not show it for the rest of the figures. Authors should show decoding accuracy changes across groups to accompany figs 4 and 5.

The decoding accuracy for the irradiated and non-irradiated groups is now shown in the consolidated Fig. 1F. We now also show the decoding accuracies for the chemogenetics experiment in Fig. S6C. Despite a trend for lower decoding accuracy the difference between CNO and baseline control recordings was not significant, which we attribute to the fact that hM4Di activation may never achieve the complete silencing of ABNs and is therefore a weaker manipulation than irradiation. However, there was a significant decrease in population Fisher information (d'^2), another measure of population information, upon silencing in ABNs (Fig. S6D). Also, single-cell Fisher information, tuning index, curve-fitting and overall activity all showed significant differences between CNO and baseline control.

- Overall, there are many instances regarding lack of detail when it comes to the methods described. For one, is it not clear what statistical analyses were used for a number of results presented. In addition, there were times when authors elected to do ANOVAs on four groups, and others where they split the groups for separate analyses. Authors should combine all groups into one plot and run appropriate ANOVAs with corrections for multiple comparisons. Additional detail is needed across the other methods as well.

We have merged some of the plots in the previous version of our manuscript (eg. the previous Fig. 1E and 2D into a combined Fig. 1F). Other plots however should be kept separate: for example it only makes sense to compare noise correlations with its own shuffled data. We only used ANOVA when comparing animals. Most of our data compares cells and we used a nested statistical approach that takes the identity of each animal into consideration. We use a non-parametric bootstrapping approach that is valid independently of whether the data is normally distributed. We are including the source code of the bootstrapping method with our analysis data. When bootstrapping is used and multiple groups are compared, we adjust p-values using the Bonferroni correction, which is a statistically conservative approach. Furthermore we consulted with the Biostatistics department at our institution (Dr. Mimi Kim) who helped us select the most appropriate statistical tests for each experiment. We are therefore very confident that we used the most appropriate statistical methods for our data. We now also take care to include in every figure legend which test of statistical significance was used.

- There were multiple instances of authors not showing the histological confirmation of sensor expression and window placements across experiments. For example, was the DCX staining in 1C from an imaged

mouse? In general, Fig 1-2 needs quantification of DCX in the in the imaged mice, with representative figures.

Our cell counts (Fig. 1D) were indeed done in the same tissue sections that were used for in vivo imaging. We have now added quantification of DCX (Fig.1D) and representative images (Fig. 1C, Fig. S2) as requested by the reviewer. We also include representative images that were all taken under the imaging window (window placement shown in low magnification images).

- No discussion was provided regarding recent work from McHugh et al., 2022 Nature Neuroscience which shows suppressing abGCs increases dentate gyrus principal firing rates. Authors should address the differences between this study and the previous.

While we did cite and discuss McHugh et al. (2022), we agree that this was very brief and have now included a more extensive discussion of our results in light of both McHugh et al. (2022) and Tuncdemir et al. (2023). There are important similarities and but also many differences between these three papers that investigated the function of ABNs in vivo. Different labs working in this field have different findings, which we believe is not unusual given that to this date very few publications report on the effect of ABNs on DG activity. We are **openly sharing the entirety of our Ca²⁺ imaging datasets, as well as our analysis pipeline**, so that the different results can more easily be reconciled (link to analysis pipeline is included, whereas full datasets will be shared upon publication).

Minor Concerns:

- Figures 1 and 2 should be combined since it seems the RC and EE groups were used in both Fig 1E and 2D.

We have combined both these figures and restructured the way we present our data.

- Figure S2 is missing scale bar, same animals irradiated or are these a different group of mice?

We've added a scale bar. These are the same mice that were imaged, which was now noted in the figure legend.

- Figure 2A/B needs better histology, IRR group looks like it has some background noise that could be masking positive staining.

We've now corrected this by ensuring that both images are displayed with same brightness and intensity levels.

- Fig 1D is not great representation of data; add compartmentalization of different textures in 1D and add legend to illustrate which traces would be considered place cells.

Done. We've re-done Fig. 1 in response to the reviewer's comments and added the calcium traces of spatially tuned and un-tuned cells to Fig 3A in the current version of the manuscript. We believe that this location makes more sense since it's where we introduce the concept of spatial tuning of individual cells.

- Figure 5A → didn't write in when they injected tamoxifen

We now corrected this oversight and updated the text so the timing of the tamoxifen injection is more clear (note that this is now Fig. 4A).

- Data from the DREADD experiment is underpowered with 2 mice in control group. IN general the numbers a low here, as other experiments are with only 4 mice.

The reviewer is correct, although we only realized this as the genotypes of animals came in after the experiment. We have now added new data from DREADD and control animals – since the experimenter was blind to the genotype we added to both groups, we now have 9 hM4Di+ mice and 5 control mice.

- Across the experiments, Authors describe the effect of single cell spatial tuning changes becoming more tuned with EE, however the authors never show the proportion of neurons that are place cells using traditional metrics, to illustrate that more GCs are spatially tuned, so the reader can compare these results to existing literature.

Our Tuning Index is a measure that is commonly used by other publications in the field and is what best allows a comparison to existing publications. We now also include the proportion of spatially tuned vs non-tuned cells, in Fig. S4C, S6A,B, as determined by curve-fitting to a Von Mises function (tuned cells have $R^2 > 0.5$), which is the method we used in the manuscript although this measure is quite variable and not quite the same as place cells. However, we should note that in our opinion there is no “traditional” metric of place cells as applied to Ca^{2+} imaging data, since the most commonly used measures were created for electrophysiology data. There are significant differences and challenges to applying the same methods to Ca^{2+} traces that have been discussed elsewhere (Climer and Dombeck, 2021; Sheintuch et al., 2022) and even if the metrics used by different labs are nominally the same, they can have different meanings. For example, some labs use the raw fluorescence traces, others detect “calcium events”, while others deconvolve their traces to obtain putative firing rates. These methodological differences can make it difficult to compare results.

In an attempt to facilitate comparisons between the results of different labs we will make our analysis code and the entirety of the Ca^{2+} data used in this paper (both unprocessed movies and calcium traces + mouse position) openly available on a repository site.

- Authors point out in page 10 that inflammation from irradiation could be an issue, a quantification of Iba1+ or other markers would help ensure this is not a major issue.

As suggested, we have now labeled the imaged tissue with an antibody against Iba1 and quantified microglial branching. A reduction in branching is generally considered to be one of the best indicators of increased inflammation (Perry et al., 2010). We found that there was a clear and significant reduction in microglial branching complexity in the irradiated tissue (Fig. S9) which is indicative of an increase in inflammation, although microglia still appeared branched and did not have a fully amoeboid phenotype. Even though it is difficult to predict the consequence of increased inflammation on information encoding, this is an important caveat to our results that we now discuss more in depth. However, our hM4Di silencing experiments recapitulate the most important effects of irradiation: decrease in single-cell activity and single-cell spatial information, as well as a decrease in goodness-of-fit and tuning curve amplitude in spatially-tuned cells. We were not able to recapitulate the population decoder experiments, possibly because hM4Di activation may never achieve the complete silencing of ABNs. Another reason is that the size of the usable field-of-view can vary from animal to animal. Some hM4Di+ mice happened to have fewer than 42 active neurons in their field-of-view and therefore could not be decoded using the same parameters used for the irradiated cohort. However, the near-perfect concordance between the data from silenced and irradiated animals reinforces our view that these effects are specific to neurogenesis.

Reviewer #2 (Remarks to the Author):

The manuscript entitled “Adult neurogenesis improves spatial information encoding in the mouse hippocampus” by Frechou and colleagues investigates the role of adult-born dentate granule cells in processing spatial information, a long-standing question with broad implications across disciplines in

neuroscience. The authors used a treadmill containing different sensory stimuli to simulate spatial processing while analyzing the activity of dentate granule cells using two-photon calcium imaging in head fixed mice. Mice were exposed to an enriched environment (EE) to increase neurogenesis and compare spatial encoding of dentate granule cells with that of regular caged mice with basal levels of adult neurogenesis, and the content of spatial information was estimated using a linear decoder algorithm. The authors first observed that decoding accuracy of granule cells was higher in mice exposed to EE than controls, suggesting that new neurons would enhance spatial encoding. Then, focal irradiation was used to prevent neurogenesis in the context of an exposure to EE, which this time resulted in no effect probably due to the lack of adult-born neurons. Finally, the involvement of new neurons in enhancing spatial information after EE was interrogated using chemogenetic inhibition of adult-born neurons using the DREADD HM4Di, which again suggested a role for adult neurogenesis. Experiments are overall well conducted and the flow of the manuscript is adequate. However, there are major experimental caveats that will need to be addressed to clarify data and conclusions.

Specific comments:

1) Fig. 2D: please explain/discuss the effect of irradiation on RC mice. Is this due to the elimination of basal neurogenesis? Can you exclude neurogenesis-unrelated effects of irradiation (i.e. inflammation)?

We cannot exclude the non-specific effects of irradiation as a potential confound in our data, as pointed out in the original version of the manuscript. We now include a careful quantification of microglia branching and found that it is less complex in irradiated animals, which is indicative of increased inflammation (Fig. S9). Even though it is difficult to predict the consequence of increased inflammation on information encoding, this is an important caveat to our results that we now discuss more extensively.

We also note that the hM4Di silencing experiments recapitulate the effects of irradiation: decrease in single-cell activity and single-cell spatial information, as well as a decrease in goodness-of-fit and tuning curve amplitude in spatially-tuned cells. We were not able to recapitulate the decrease in population decoder accuracy, possibly because hM4Di activation may never achieve the complete silencing of ABNs.

2) In several plots along the manuscript, mean \pm SEM bars seem out of place. See, for instance, Fig. 3E, where all bars seem too high compared to the scatter density. Please verify and correct all plots.

We have verified these and believe that they are correct.

3) It is very important to estimate and discuss and analyze the data in light of the putative age of the neuronal populations being affected by the experimental manipulations. This is a critical point in the manuscript.

We agree with the reviewer's feedback and we now clarify that all our manipulations are primarily targeting a cohort of ABNs that is 4-6 weeks post-mitosis.

a) Irradiation; mice were irradiated 4 weeks before the exposure to EE, and 7.5 weeks before behavior/imaging experiments. Thus, these mice would entirely lack adult neurogenesis and miss all adult-born neurons up to 7.5 weeks of age.

This is correct. Irradiation will permanently ablate adult neurogenesis and there will be no ABNs younger than 7.5 weeks of age.

b) EE exposure: EE is known to enhance survival of immature neurons. What would be the age of these neurons at the time when recordings were done? Ideally, this age should be measured, but it might be inferred if one would consider that EE would act when neurons are around 2-3 weeks old. Populations enhanced by 2 weeks of EE would perhaps be around 5 to 7 weeks old.

Enriched environments with running wheels increase neuronal population and survival, therefore acting on a relatively wide range of progenitors/neurons. However, the largest effect on in the survival of neurons that are ~7 to 21 days post-mitosis. Therefore the bulk of the EE-enhanced population will be ~4.5 – 6.5 weeks old at imaging (Tashiro et al., 2007). This is the developmental stage when ABNs are thought to make the biggest impact on DG networks and on cognition (Denny et al., 2012; Gu et al., 2012), and also the age at which we silenced ABNs (Fig. 4 of the revised manuscript).

c) HM4Di silencing: in Fig. 5, *Ascl1-CreERT2/flox-hM4D* mice were TAM-induced at 8 weeks of age, labeling neurons born at around right this time (0 days old). Mice were exposed to EE simultaneously (presumably enhancing the 2-3 w neuronal population?). Thus, CNO would be silencing neurons between 12 to 27 days, whereas EE would be enhancing the 5-7 weeks population (but not the cohort expressing HM4D).

As pointed out by another reviewer, the timing of the Tamoxifen was unclear in Fig. 5 and in the text of the original manuscript. Tamoxifen is injected 2 weeks before surgery (now shown in corrected figure), meaning that the oldest hM4Di+ ABNs are ~5.5 weeks old at the time of imaging/silencing. Since *Ascl1* is expressed in transient amplifying progenitors and in a subset of Type 1 adult neural stem cells, some hM4Di neurons will continue to be generated for a short time after Tamoxifen injection, meaning that they are slightly younger than 5.5 weeks (Pilz et al., 2018) **This means that silenced neurons will be ~4-6 weeks old, which is the same population that is enriched in the EE group** and also the developmental stage when ABNs are thought to make the biggest impact on DG networks and on cognition (Denny et al., 2012; Gu et al., 2012).

CNO treatment produced, in general, very limited effects when compared to focal irradiation (see for example, Fig. 5M and compare to 4F). CNO would only be silencing neurons up to 27 days, whereas irradiation would be eliminating all new neurons. Thus, these different experimental designs interrogate neuronal non-overlapping populations of granule cells. To target roughly similar populations and attribute the lack of EE effect in IRR to neurogenesis, chemogenetic silencing should be directed to that same neuronal population (5-7 weeks old). Perhaps under those conditions the effect of CNO on spatial encoding would be more robust and reliable.

We now clarify that our different approaches target the same neuronal population: ABNs that are ~4-6 weeks post-mitosis. We think that out that our chemogenetic experiments fully recapitulate the most relevant findings of the irradiation experiments: decrease in single-cell activity and single-cell spatial information, as well as a decrease in goodness-of-fit and tuning curve amplitude in spatially-tuned cells. We were not able to recapitulate the population decoder experiments, possibly because hM4Di activation may never achieve the complete silencing of ABNs.

Reviewer #3 (Remarks to the Author):

Frechou and colleagues report a study describing how neurogenesis induced by an enriched environment can contribute to the coding of spatial changes in a mouse model. Highly clinically relevant results were obtained using state-of-the-art techniques.

The authors know that a radiation dose of 10 Gy induces inflammation. Their protocol includes a month-long recovery period before imaging sessions, which they claim would significantly decrease inflammation. To validate this hypothesis, the level of radiation-induced inflammatory markers (such as IL-1 β , IL-6, TNF α) could have been measured in additional groups before irradiation, 6 hours after irradiation and 30 days later. On the other hand, the reviewer agrees that their acute chemogenetic silencing of ABNs supports results obtained after ablating by radiation the neurogenesis in dentate gyrus. Irradiation of a specific and small area in the mouse brain is difficult. A state-of-the-art animal irradiator was used to delivery a bilateral radiation dose. Consequently, a significant radiation dose only 20% to 30% lower than delivered in the dentate gyrus was deposited in the surrounding brain areas. The level of

damage in these could have been assessed by histopathology. A stereotactic irradiation of the radiation dose could have been explored.

We agree with Reviewer 3 that the high concordance between the chemogenetic silencing and irradiation experiments supports the conclusions of our manuscript: in both cohorts we saw a decrease in single-cell activity and single-cell spatial information, as well as a decrease in goodness-of-fit and tuning curve amplitude in spatially-tuned cells. As suggested by Reviewer 3 and by the other reviewers, we have now labeled the imaged tissue with an antibody against Iba1 and quantified microglial branching, which is thought to be a good indicator of inflammation (Perry et al., 2010). We found that there was a clear and significant reduction in microglial branching complexity in the irradiated tissue (Fig. S9) which is indicative of an increase in inflammation, although microglia still appeared branched and did not have a fully amoeboid phenotype. Even though it is difficult to predict the consequence of increased inflammation on information encoding, this is an important caveat to our results that we now discuss more in depth.

Minor comments

1) Mention in the captions the length corresponding to the white bar added to the figure, and the meaning of the star “*, **, ***”.

Done.

2) Figure 2: Authors are encouraged to add arrows to indicate where DCX positive cells are located.

This figure was combined with Fig. 1 as suggested by Reviewer 1, and we added arrows to denote DCX+ cells in the dorsal and ventral leaves of the DG as requested by Reviewer 3.

Bibliography

Carrier-Ruiz A, Sugaya Y, Kumar D, Vergara P, Koyanagi I, Srinivasan S, Naoi T, Kano M, Sakaguchi M (2021) Calcium imaging of adult-born neurons in freely moving mice. STAR Protoc 2:100238.

Chen T-W, Wardill TJ, Sun Y, Pulver SR, Renninger SL, Baohan A, Schreiter ER, Kerr RA, Orger MB, Jayaraman V, Looger LL, Svoboda K, Kim DS (2013) Ultrasensitive fluorescent proteins for imaging neuronal activity. Nature 499:295–300.

Climer JR, Dombeck DA (2021) Information Theoretic Approaches to Deciphering the Neural Code with Functional Fluorescence Imaging. eNeuro 8:ENEURO.0266-21.2021.

Danielson NB, Kaifosh P, Zaremba JD, Lovett-Barron M, Tsai J, Denny CA, Balough EM, Goldberg AR, Drew LJ, Hen R, Losonczy A, Kheirbek MA (2016) Distinct Contribution of Adult-Born Hippocampal Granule Cells to Context Encoding. Neuron 90:101–112.

Denny CA, Burghardt NS, Schachter DM, Hen R, Drew MR (2012) 4- to 6-week-old adult-born hippocampal neurons influence novelty-evoked exploration and contextual fear conditioning. Hippocampus 22:1188–1201.

Gu Y, Arruda-Carvalho M, Wang J, Janoschka SR, Josselyn SA, Frankland PW, Ge S (2012) Optical controlling reveals time-dependent roles for adult-born dentate granule cells. Nat Neurosci 15:1700–1706.

Johnston S, Parylak SL, Kim S, Mac N, Lim C, Gallina I, Bloyd C, Newberry A, Saavedra CD, Novak O, Gonçalves JT, Gage FH, Shtrahman M (2021) AAV ablates neurogenesis in the adult murine hippocampus. Elife 10:e59291.

Kumar D et al. (2020) Sparse Activity of Hippocampal Adult-Born Neurons during REM Sleep Is Necessary for Memory Consolidation. Neuron 107:552-565.e10.

McHugh SB, Lopes-dos-Santos V, Gava GP, Hartwich K, Tam SKE, Bannerman DM, Dupret D (2022) Adult-born dentate granule cells promote hippocampal population sparsity. Nat Neurosci 25:1481–1491.

- Perry VH, Nicoll JAR, Holmes C (2010) Microglia in neurodegenerative disease. *Nat Rev Neurol* 6:193–201.
- Pilz G-A, Bottes S, Betizeau M, Jörg DJ, Carta S, Simons BD, Helmchen F, Jessberger S (2018) Live imaging of neurogenesis in the adult mouse hippocampus. *Science* 359:658–662.
- Pilz G-A, Carta S, Stäuble A, Ayaz A, Jessberger S, Helmchen F (2016) Functional Imaging of Dentate Granule Cells in the Adult Mouse Hippocampus. *J Neurosci* 36:7407–7414.
- Sheintuch L, Rubin A, Ziv Y (2022) Bias-free estimation of information content in temporally sparse neuronal activity. *PLoS Comput Biol* 18:e1009832.
- Tashiro A, Makino H, Gage FH (2007) Experience-Specific Functional Modification of the Dentate Gyrus through Adult Neurogenesis: A Critical Period during an Immature Stage. *J Neurosci* 27:3252–3259.
- Tuncdemir SN, Grosmark AD, Chung H, Luna VM, Lacefield CO, Losonczy A, Hen R (2023) Adult-born granule cells facilitate remapping of spatial and non-spatial representations in the dentate gyrus. *Neuron*:S0896-6273(23)00703-1.
- Tuncdemir SN, Grosmark AD, Turi GF, Shank A, Bowler JC, Ordek G, Losonczy A, Hen R, Lacefield CO (2022) Parallel processing of sensory cue and spatial information in the dentate gyrus. *Cell Rep* 38:110257.

REVIEWER COMMENTS

Reviewer #1 (Remarks to the Author):

In the revised manuscript, the authors have addressed my initial concerns about the clarity and detail in the presentation. However, there is one minor issue that remains around the status of neurogenesis in the control mice. The authors say that indeed, past experiments are correct, AAV kills all young newborn neurons, and that in their experiments the only adult generated neurons that remain are ones that escaped infection due to not being in the direct injection site. While that may be the case, it is important for readers to see this convincingly with quantification. In the revised manuscript Figure S2, the number of DCX neurons is lower than that shown in figure 1C, and no quantification is provided for how many DCX cells express jRGECO (is it zero?). In addition, where the DCX cells that escaped cell death via AAV reside relative to the imaged portion of DG should be shown.

Reviewer #2 (Remarks to the Author):

The authors have addressed my concerns in full. The manuscript is now suitable for publication.

Reviewer #3 (Remarks to the Author):

The authors responded well to my requests for corrections and modifications.

We sincerely thank the reviewers for their careful reviews. We were happy to note that Reviewers 2 and 3 are in support of publication. Here we address the minor issues raised by Reviewer 1.

Reviewer #1 (Remarks to the Author):

In the revised manuscript, the authors have addressed my initial concerns about the clarity and detail in the presentation. However, there is one minor issue that remains around the status of neurogenesis in the control mice. The authors say that indeed, past experiments are correct, AAV kills all young newborn neurons, and that in their experiments the only adult generated neurons that remain are ones that escaped infection due to not being in the direct injection site. While that may be the case, it is important for readers to see this convincingly with quantification. In the revised manuscript Figure S2, the number of DCX neurons is lower than that shown in figure 1C, and no quantification is provided for how many DCX cells express jRGECO (is it zero?). In addition, where the DCX cells that escaped cell death via AAV reside relative to the imaged portion of DG should be shown.

1. Quantification of DCX and jRGECO1a overlap

We collected high magnification (1.0 Airy) z-stacks to verify whether there is any colocalization between DCX and jRGECO1a. We now imaged 52 DCX-expressing cell somas in 3 sections from 3 different mice, using the same settings as in Supplementary Figure 2b. We found that there is **no colocalization (i.e. zero overlap)** between DCX and jRGECO1a. **This was now noted in the legend of Supplementary Fig. 2.**

2. Differences in number of DCX-expressing cells between Fig. 1C and Supplementary Fig. 2

The numbers of DCX-expressing cells can be very variable from one section to another or even within the same section depending on the extent of jRGECO1a expression (see arrowhead in Supplementary Fig. 2a). This explains the differences between Supplementary Fig. 2 and Fig. 1C. Both images were acquired in the center region of the implant but Supplementary Fig. 2 images were acquired closer to the epicenter of the viral injection, as we needed an area with strong jRGECO1a to quantify the overlap with DCX.

3. Location of window implant relative to DCX-expressing cells

Fig. R1 Moving the objective off-center will result in a reduction of the effective numerical aperture, thus limiting the area that can be imaged. This figure is not included in the manuscript.

As the implant is centered on the AAV injection site, the location where calcium imaging movies are acquired will generally contain fewer DCX-expressing cells than regions that are distant from the injection. However, the pattern of DCX expression is highly variable across mice, depending primarily on the spread of the virus during injection. For example, in some animals the virus will strongly label one leaf of the DG but not the other, or even superficial cells but not the area adjacent to the hilus.

The tissue sections shown in the manuscript were taken from the central area of the window implant (see implant location in Supplementary Fig. 2a). We are sure that this is the region that is imaged, since moving the objective off-center by ~300 μm or more will clip the excitation and emission light (Fig. R1), thus reducing the effective numerical aperture, resulting in degraded image quality. Nevertheless, we cannot match the exact location of imaging for each animal, as the tissue has already been sectioned coronally.

We have now added a note to the results section indicating that the region that was imaged has a reduced number of DCX-expressing cells.

REVIEWERS' COMMENTS

Reviewer #1 (Remarks to the Author):

The response to the previous round of comments was satisfactory.